# Scalable In-context Ranking with Generative Models

**Nilesh Gupta**[1]    **Chong You**[3]    **Srinadh Bhojanapalli**[3]    **Sanjiv Kumar**[3]
**Inderjit Dhillon**[1,2]    **Felix Yu**[3]

[1]University of Texas at Austin    [2] Google    [3]Google Deepmind
 https://github.com/nilesh2797/BlockRank

## Abstract

In-context Ranking (ICR) is an emerging paradigm for Information Retrieval (IR), which leverages contextual understanding of LLMs by directly incorporating the task description, candidate documents, and the query into the model's input prompt and tasking the LLM to identify relevant document(s). While it is effective, efficiency is a significant challenge in this paradigm, especially as the candidate list grows due to quadratic / super-linear scaling of attention operation with context length. To this end, this paper first identifies inherent and exploitable structures in the attention of LLMs finetuned for ICR: (1) *inter-document block sparsity* – attention is dense within each document block but sparse across different documents in the context; and (2) *query-document block relevance* – the attention scores from certain query tokens to a document block in middle layers strongly correlate with that document's actual relevance. Motivated by these observations, we introduce BlockRank (Blockwise In-context Ranking), a novel method that adapts the attention operation in an LLM by (a) architecturally enforcing the observed inter-document block sparsity, reducing attention complexity from quadratic to linear without loss in performance, and (b) optimizing query-document block relevance for true relevant documents during fine-tuning using an auxiliary contrastive training objective, improving retrieval in attention. Experiments on BEIR, MSMarco and NQ with Mistral-7B demonstrate that BlockRank Mistral matches or outperforms existing SOTA listwise rankers and controlled fine-tuned baseline while being significantly more efficient at inference ($4.7\times$ for 100 MSMarco documents in context) and scaling gracefully to long-context shortlists - around 500 documents in-context ($\sim 100K$ context length) within a second, presenting a scalable and effective solution for ICR.

## 1   Introduction

Information retrieval (IR) is the problem of finding relevant content from a large document corpora. While sparse retrieval methods based on word-level matching have existed for decades [Robertson et al., 2009, Formal et al., 2021], modern IR systems increasingly leverage deep neutral network based representations, which achieve their success through a superior ability to capture deep semantic relationships [Karpukhin et al., 2020]. Recently, generative large language models (LLMs) [Team et al., 2023, Achiam et al., 2023] have emerged as a revolutionary paradigm that transforms many sub-fields of machine learning, including IR. Through pre-training on the web, LLMs absorb an enormous amount of world knowledge and demonstrate remarkable capabilities in dialogue, question answering, reasoning, and beyond [Wei et al., 2022].

The powerful capabilities of LLM open up novel approaches for IR as well. One emerging paradigm is the In-context Ranking (ICR) [Lee et al., 2024, Ma et al., 2023], which directly leverages an LLM's contextual understanding capabilities. In this setup, a query and a list of candidate documents are

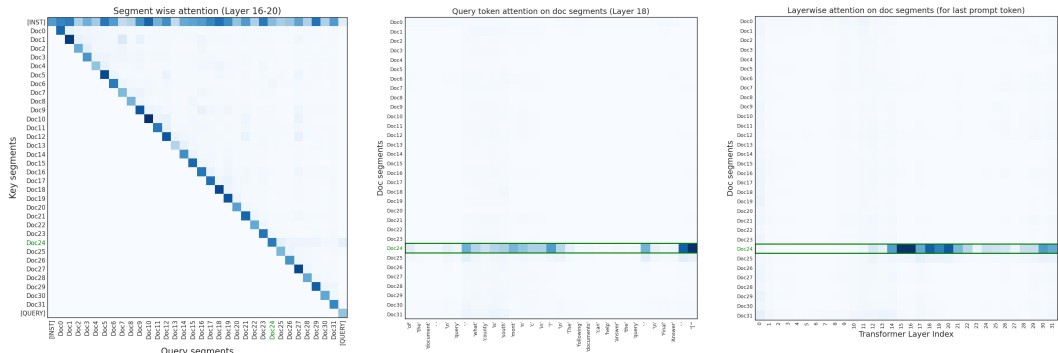

Figure 1: Analysis of attention patterns in Mistral-7B performing In-context Ranking (ICR) on MSMarco. (*left*) Attention averaged over middle layers 16-21 reveals structural sparsity — a strong diagonal (intra-document attention needed for local context processing) and significant attention to the first row (focus on the query-based instruction). (*middle*) Attention in Layer 18 from individual query tokens to document segments. Certain tokens (the last token, ':') attend primarily to the relevant document only (i.e., Doc24, highlighted in green). (*right*) Attention from final query tokens across layers shows retrieval signals strengthening in middle layers. These patterns motivate our BlockRank approach.

formatted together within the LLM's input prompt (see Figure 3), tasking the model to identify the most relevant document(s), often through the generative decoding process. ICR holds the promise of considering the query and all candidates simultaneously while performing relevance judgements.

Despite this promise, LLM-based ICR introduces significant efficiency challenges. As the number of candidate documents increases, the input context length grows rapidly, making inference computationally expensive and memory-intensive, due to quadratic/super-linear complexity of the attention mechanism. Current methods [Lee et al., 2024, Sun et al., 2023, Pradeep et al., 2023a,b] typically treat the LLM as a black-box or do not fully utilize the structure of the ICR task i.e. the input prompt is composed of a sequence of potentially independent candidate documents conditioned on a shared instruction prompt. Moreover, as we discuss in Section 5.3 and Section D.1 of Appendix, auto-regressive decoding is not best suited for this task when decoding multiple predictions from the fine-tuned model (see Table 4).

**Paper Contributions.** To this end, we first investigate how standard LLMs process information within the specific task structure. We conduct an analysis of the attention patterns of a fine-tuned Mistral-7B model when prompted on ICR examples derived from MSMarco dataset (see Section 3 for details and Figure 1 for visualizations). This analysis reveals two structural properties: (1) *inter-document block sparsity* – most document tokens focusing locally (primarily within their own document, on instructions, or one or two other documents), rather than attending densely across all candidate documents. (2) *query-document block relevance* – similar to the findings of Wu et al. [2024], Chen et al. [2025], we find that last and some specific query tokens like ":" (that signal start of the potential document generation process) develop strong attention weights towards relevant document tokens, particularly in the model's middle layers.

Building up on these insights, we propose BlockRank (Blockwise In-context Ranking), an efficient in-context ranking method. BlockRank introduces two modifications (visualized in Figure 2) to standard LLM architecture and fine-tuning: (1) architecturally, it imposes a structured sparse attention in which document tokens attend only causally to their own content and shared instruction tokens, reducing attention complexity from quadratic to linear; and (2) it incorporates a contrastive learning objective to explicitly optimize internal attention from signal-carrying query tokens toward relevant documents, which helps the BlockRank model in two fronts: (a) attend strongly to the relevant document in context, improving retrieval quality (see Table 3); (b) the ability to reliably infer relevance based on attention concentration during the prefill stage, leading to further speedups in inference compared to iterative decoding (see Section 4.3). To summarize, the main contributions of this work are:

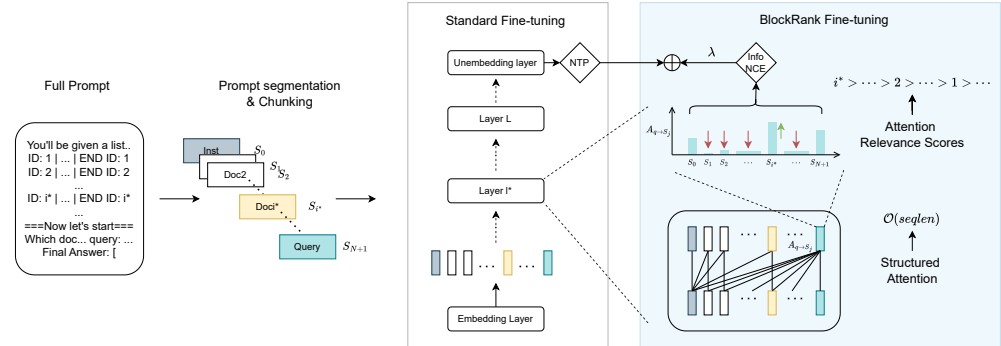

Figure 2: BlockRank starts with chunking the full prompt into segments and then processes it using structured attention, where the documents only attend to themselves and the instruction segment, while the query segment attends to the full prompt. It also incorporates an auxiliary attention loss ($\mathcal{L}_{\text{aux}}$) from a middle layer ($l^*$) that increases sharpness of attention on the relevant documents and enables an alternate inference mechanism using attention scores derived from $l^*$.

- an analysis that characterizes attention patterns in LLMs fine-tuned for ICR, identifying key sparsity structures and latent retrieval signal carriers (specific query tokens in middle layers).
- an efficient approach BlockRank for In-context Ranking that enforces a structured sparse attention and a contrastive training objective on internal attention.
- extensive experiments on standard retrieval benchmarks (BEIR, MSMarco and NQ) demonstrating that BlockRank achieve strong ICR performance, matching or outperforming strong baselines as well as full fine-tuned model (see Table 1, 2), while being order of magnitude efficient at inference (see Figure 4).

The remainder of this paper is organized as follows: Section 2 describes the problem setup and discusses related work. Section 3 details our analysis of LLM attention in ICR. Section 4 presents the BlockRank methodology. Section 5 reports experimental results, and Section 6 concludes the paper.

## 2 Problem Setup and Related Work

This section formally defines the ICR task addressed in this paper. We also review relevant prior work that uses LLM for IR, and position our BlockRank method in the context of the literature.

### 2.1 Problem Formulation: In-context Ranking

Given a collection of $n$ documents $\mathcal{D} = \{d_1, \ldots, d_n\}$ and a query $q$, the goal of IR is to return a subset of $\mathcal{D}$ that are *relevant* to $q$. In this paper, we consider documents and queries in the form of text that can be parsed by an LLM, though the discussion may also apply to visual and audio data when the LLM is multi-modal. Following standard practices [Lee et al., 2024], we define an ICR prompt as a composition of the list of documents together with the query as following:

$$\text{prompt}(q, \mathcal{D}) \doteq \text{``\{Inst\}.\{d_1\}, \ldots, \{d_N\}.\{q\}''} \tag{1}$$

In practice, processing the entire corpus $\mathcal{D}$ (where $n$ can be millions) within a single prompt is infeasible due to LLM context length limitations. Therefore, the ICR task we consider in this paper operates on a smaller candidate list $\mathcal{D}_q = (d_1, \ldots, d_N) \subseteq \mathcal{D}$, where $N$ is the number of candidates ($N = \mathcal{O}(100)$ in our experiments) retrieved by a first-stage retrieval model (e.g. dual-encoder). The prompt in (1) is thus applied to this candidate list $\mathcal{D}_q$. Furthermore, each document representation $\{d_i\}$ within the prompt often includes structured formatting beyond just the raw text $c_i$, such as its unique identifier $id_i$. We adopt the format from [Lee et al., 2024], also illustrated in Figure 3, explicitly demarcating document start, content, and end, along with identifiers (e.g., `ID:` $id_i$ `|` `CONTENT:` $c_i$ `|` `END ID:` $id_i$).

*Inst* is a description of the retrieval task and can also include the query $q$. While excluding the query from the instruction prefix is desirable from an efficiency standpoint – as it would allow for the query-independent representations of documents to be pre-processed and cached offline – we find this leads to a noticable drop in performance in our experiments (see Table 6). We hypothesize,

```
You will be given a query and a list of documents. Each document will be formatted as ID: <id> |
CONTENT: <content> | END ID: <id>. You need to read carefully and understand all of them. The query
is: which classification group contains the most organisms, and your goal is to find all document(s)
from the list that can help answer the query.
```

```
ID: 1 | CONTENT: This is a diverse group of organisms. It includes plants, animals.. | END ID: 1

...  [Documents in-between omitted for brevity] ...

ID: N | CONTENT: Organisms composed of eukaryotic cells are divided into 4 main.. | END ID: N
```

```
====== Now let's start!  ======
Which document is most relevant to answer the query? Print out the ID of the document.  Query: which
classification group contains the most organisms.  The following document(s) can help answer the query:
```

```
Final Answer:  ['20']
```

Figure 3: Example structure of the prompt template used in our experiments, showing query-based instruction, abbreviated document list, and the final query section.

including the query in *Inst* allows the model to condition each document's representation on the specific information need from the outset, enabling it to better focus on query-relevant facts and signals within each document during processing. We note that existing listwise LLM re-rankers [Sun et al., 2023, Pradeep et al., 2023b] also apply a similar formatting where the query appears before the documents. Our preliminary experiments show that one can replace the query with a similar-looking document from the corpus, suggesting that future work can potentially explore conditioning document representations within clusters to alleviate the need for query-dependent processing.

The objective of *In-context Ranking* is then formally defined as follows: given the $\mathrm{prompt}(q, \mathcal{D}_q)$ constructed from the query $q$ and the candidate list $\mathcal{D}_q$, train or utilize an LLM $f_\theta$, with $\theta$ being the model weight, to effectively identify and output the identifiers $id^*$ corresponding to the $d^* \in \mathcal{D}_q$ deemed relevant to $q$. Typically it is achieved by predicting a ranked permutation of $\mathcal{D}_q$ and taking the top $k$ elements. The central challenge addressed in this paper is to develop methods to train $f_\theta$ and perform this prediction both effectively (high accuracy) and efficiently (low computational cost), particularly as the candidate list size $N$ increases.

## 2.2 Related Work

Our work builds upon research in neural retrieval, the rapidly evolving field of using LLMs for IR, and efficient attention mechanisms.

**Neural Re-ranking and Retrieval Models.** Prior to LLMs, neural information retrieval saw significant progress with methods like dense dual-encoder retrieval (e.g., DPR [Karpukhin et al., 2020], ANCE [Xiong et al., 2020]) offering efficient first-stage filtering, and cross-encoder models (e.g., monoBERT [Nogueira and Cho, 2020], monoT5 [Nogueira et al., 2020]) providing high re-ranking effectiveness through deep query-document interaction. Late interaction models like ColBERTv2 [Santhanam et al., 2022] aimed to balance the trade-off between dual and cross encoders. *Our work is architecturally distinct from traditional neural IR methods* as it operates within the in-context ranking paradigm, where a single LLM processes the query and the entire candidate list simultaneously in one context window allowing full contextualization (query and output representations are conditioned on the full set of candidate documents) and complex instruction-following (e.g., "Find documents that disagree with...").

**LLMs as Listwise Re-rankers.** The ability of LLMs to process and reason over long contexts spurred their application to listwise re-ranking [Ma et al., 2023] (or In-context Ranking), where multiple candidates are processed simultaneously. Initial successes often involved prompting large proprietary models like GPT-3.5/4 [Sun et al., 2023] in zero-shot or few-shot settings. While effective, these approaches typically incur high computational costs and often rely on auto-regressive generation to output rankings or relevance scores, adding latency. More recent work focuses on adapting open-source LLMs (e.g., Llama, Mistral, Zephyr, Vicuna) for this task [Pradeep et al., 2023b,a, Zhang

et al., 2023] and improving efficiency, for instance using Seq2Seq architectures [Tamber et al., 2023] and using single-token decoding [Reddy et al., 2024]. Recent papers have also shown the existence of retrieval heads (attention heads that carry strong retrieval signals) in many modern LLMs [Wu et al., 2024] and their usefulness in inferring retrieval signals [Chen et al., 2025]. *Our work differs from these methods by introducing effective task specific restructuring of the attention architecture for efficiency and an explicit fine-tuning objective to directly train the model's attention patterns for the ranking task.*

**In-context Retrieval / Ranking** Lee et al. [2024] studied the In-context Retrieval (ICR) paradigm for various frontier LLMs, demonstrating that long-context models can match the performance of specialized retrieval systems when processing corpora of up to a few thousand documents. Our work builds on this paradigm but addresses a more challenging setting. The evaluation in that study is performed on a random subset of documents from the full corpus. In contrast, *our experiments focus on ranking the top-k hard candidates* returned by a strong first-stage retriever i.e. In-context Ranking task. This task is arguably more difficult, as the model must distinguish between many semantically similar documents to identify the correct answer. We argue that this hard negative setting is a more faithful simulation of the practical application of LLMs in retrieval pipelines. Moreover, processing long lists of documents within the LLM context remains challenging, with studies highlighting difficulties in effectively utilizing long-range information [Goldman et al., 2024].

**Efficient and Structured Attention.** The quadratic complexity of the standard self-attention has spurred extensive research into more efficient attention approximations. Many successful approaches enforce a **structured sparsity** on the attention matrix, reducing complexity from quadratic to sub-linear. Notable examples include methods based on sliding windows (e.g., Longformer [Beltagy et al., 2020]), global-local patterns (e.g., BigBird [Zaheer et al., 2020]), and other block-wise structures. While these methods are designed for general long-context processing, *BlockRank's structured attention can be seen as a task-specific instance of this paradigm.* BlockRank's sparsity is semantically informed by the logical structure of the in-context ranking task itself—separating instructions, documents, and the query. This content-aware structuring allows for a highly efficient architecture tailored for ICR.

## 3 Emergence of Structured Attention in In-context Ranking

Before introducing our method, we first analyze the attention mechanisms of a standard LLM when performing the ICR task defined in Section 2.1. The analysis below is anchored on Figure 1 which is based on a random sample, we provide more results in the Appendix Section D.

**Analysis Setup.** We conduct our analysis using a Mistral-7B-v0.3 model [Jiang et al., 2023] fine-tuned on the ICR task with data derived from MSMarco (as described in Section 5.1). In particular, our fine-tuning objective is the standard Next Token Prediction (NTP) loss, without any modifications. We feed this model $\text{prompt}(q, \mathcal{D}_q)$. Let the resulting input token sequence be $T = (t_1, \ldots, t_L)$.

Our analysis focuses on the attention probabilities computed within the transformer layers. Given a layer index $l \in \{1, \ldots, L_{model}\}$ where $L_{model}$ is the total number of layers, and an attention head index $h$, we denote the attention probability from a query token $t_i$ to a key/value token $t_j$ as $\alpha_{ij}^{(l,h)}$. We often consider the attention averaged across all $H$ heads in a layer: $\alpha_{ij}^{(l)} = \frac{1}{H} \sum_{h=1}^{H} \alpha_{ij}^{(l,h)}$. We examine interactions between different types of tokens by partitioning the token indices $\{1, \ldots, L\}$ into sets corresponding to the instructions ($T_{Inst}$), the query ($T_q$), and each document ($T_{d_k}$ for $k \in \{1, \ldots, N\}$). We visualize these interactions using heatmaps, with representative examples shown in Figure 1.

**Observation 1: Inter-document Block Sparsity** Our first key observation is that the attention patterns exhibited by document tokens are structured and sparse, rather than uniformly dense. This is clearly visible in Figure 1(a), which shows the segment-wise attention in the middle layers. The heatmap is dominated by the diagonal, indicating strong *intra-document attention*: for a token $t_i \in T_{d_k}$, the sum of attention probabilities towards other tokens within the same document, $\sum_{t_j \in T_{d_k}} \alpha_{ij}^{(l)}$, is significantly higher than attention towards other parts of the context.

This observed structured sparsity implies that computing full attention matrix might be largely redundant for this task. A significant portion of the computation could potentially be saved by

enforcing an attention pattern that focuses on local (intra-document) and instructional context, directly motivating the structured sparse attention employed in BlockRank.

**Observation 2: Query-document Block Relevance**   Our second key observation is that certain tokens within the query $T_q$ attends primarily to relevant documents only, particularly in the middle layers of the transformer.

Figure 1(b) illustrates this at Layer 18. It maps the attention from individual query tokens (x-axis) to the different document segments (y-axis). We observe that certain tokens, such as delimiters (`':'`) and end of prompt tokens, exhibit distinct sharp attention distributions. These specific "signal carrier" tokens attend more strongly towards the segment corresponding to the ground-truth relevant document $d^*$ (i.e., Doc24, highlighted in the figure) compared to irrelevant documents $d_k$ ($k \neq *$). Formally, let $A_{i \to d_k}^{(l)} = \sum_{t_j \in T_{d_k}} \alpha_{ij}^{(l)}$ be the total attention from query token $t_i$ to document $d_k$ at layer $l$. For specific $t_i \in T_q$ identified as signal carriers and middle layers $l$, we observe $A_{i \to d^*}^{(l)} > A_{i \to d_k}^{(l)}$ for most $k \neq *$. We hypothesize that such structural tokens carry strong retrieval signals as they often precede $T_{d^*}$ (by design during fine-tuning but also during pre-training), hence their attention gets biased towards the in-context $d^*$ segment in order to predict the succeeding $T_{d^*}$ tokens.

Furthermore, the layer depth plays a critical role in the emergence of these signals. Figure 1(c) tracks the attention $A_{i \to d_k}^{(l)}$ from final query tokens $t_i$ to all document segments $d_k$ across all layers $l \in \{1, \ldots, L_{model}\}$. The plot shows that the discriminative signal is weak in the initial layers, emerges and strengthens significantly in the middle layers (approximately layers 8-24), and persists or slightly diffuses in the final layers.

# 4   BlockRank: Blockwise In-context Ranking

Motivated by the attention analysis presented in Section 3, we propose Blockwise In-context Ranking (BlockRank), an efficient in-context ranking method. BlockRank comprises of following components (see Figure 2): a structured attention mechanism enforcing sparsity, an auxiliary attention loss to enhance retrieval signals in attention operation, and an alternative attention-based inference method. We detail each component below.

## 4.1   Blockwise Structured Attention

The core of BlockRank's efficiency during fine-tuning and inference stems from restructuring of attention mechanism designed to enforce the sparse patterns observed in Section 3.

**Enforcing inter-document block sparsity.**   we modify attention operation such that:

- **Document Tokens** ($t_i \in T_{d_k}$ for $k \in \{1, \ldots, N\}$): only attend to tokens within their own document chunk ($t_j \in T_{d_k}$) and tokens within the instruction chunk ($t_j \in T_{Inst}$).
- **Query Tokens** ($t_i \in T_q$): attend to all tokens in the prompt ($t_j \in T = \cup_k T_k$) to gather context for identifying the relevant document(s).
- **Instruction Tokens** ($t_i \in T_{Inst}$): attend causally within the instruction segment itself.

Instead of constructing large, explicit sparse attention masks, we implement this structured attention efficiently using the chunked representation defined as follows: the long prompt is first segmented into its logical components $S_0 = Inst$, $S_k = d_k$ for $k \in \{1, \ldots, N\}$, and $S_{N+1} = q$. Each segment $S_k$ is then processed (via standard sequence length chunking or padding) to form fixed-length chunks, typically of length $L_{chunk}$ tokens. Let the token sequence corresponding to chunk $S_k$ be $T_k \subset T = (t_1, \ldots, t_L)$, where $T$ is the (potentially virtual) concatenation of all chunk sequences.

Each chunk $S_k$ can be processed largely in parallel (e.g., distributed along the batch dimension). Let $Q_k^{(l)}, K_k^{(l)}, V_k^{(l)}$ be the query, key, and value matrices for chunk $S_k$ at layer $l$. The attention output for a token $t_i$ in chunk $S_k$ is computed as follows:

- If $S_k$ is a document chunk ($k \in \{1, \ldots, N\}$): The attention output is computed using self-attention within the chunk and cross-attention only to the keys and values from the instruction chunk: $Attention(Q_k^{(l)}, [K_k^{(l)}, K_{Inst}^{(l)}], [V_k^{(l)}, V_{Inst}^{(l)}])$. Attention to other document chunks $S_{m \neq k}$ and the query chunk $S_q$ is effectively masked out.

- If $S_k$ is the query chunk ($k = N + 1$): The attention output is computed using self-attention within the chunk and cross-attention to the keys and values from *all* other chunks:
  $Attention(Q_q^{(l)}, [K_q^{(l)}, K_{Inst}^{(l)}, K_{d_1}^{(l)},$
  $\ldots, K_{d_N}^{(l)}], [V_q^{(l)}, V_{Inst}^{(l)}, V_{d_1}^{(l)}, \ldots, V_{d_N}^{(l)}]).$
- Instruction chunk attention ($k = 0$) is standard causal self-attention.

This computes only the necessary attention scores, drastically reducing the computational cost, converting quadratic attention to linear. Please see Appendix Section C for more details and complexity analysis.

**Permutation-invariant Position Embedding.** To complement the structured attention, we employ a specialized position embedding that reinforces the logical separation of the prompt's components. This also helps the model learn position-invariant representations for documents [Tang et al., 2023] and distinguish the query's unique role. Specifically, tokens in the instruction segment ($T_{Inst}$) are assigned standard sequential positions starting from 0. For all document segments ($T_{d_k}$), we use a shared local position space. Each document's tokens are assigned positions beginning immediately after the instruction segment, as if it were the only document present. For example, if the instruction has length $L_{Inst}$, the first token of *every* document $d_k$ is assigned the position $L_{Inst}$. This encourages the model to apply a consistent, order-invariant function to each document, mitigating any bias from its absolute position in the candidate list. Finally, to distinctly separate the query from the document corpus, its tokens ($T_q$) are assigned positions starting from a large, fixed offset. In our experiments, we use an offset of 8192, so the query tokens receive positions $[8192, 8193, \ldots]$. This large gap ensures that the relative positional encodings between any query token and any document token are significantly different.

## 4.2 Auxiliary Attention Loss ($\mathcal{L}_{\text{aux}}$)

To explicitly optimize query-document block relevance for relevant documents during fine-tuning, we introduce an auxiliary loss $\mathcal{L}_{\text{aux}}$ applied at a specific middle layer $l^*$ (determined empirically, see Section D.3 in the Appendix). This loss encourages "signal-carrier" query tokens to attend more strongly to the relevant document.

More specifically, let $T_{q,signal} \subset T_q$ be the set of indices for the identified signal-carrying query tokens. Based on our prompt template and empirical analysis we set $T_{q,signal} = [\text{":"}, \text{"[ '"}]$. Let $T_{docs} = \bigcup_{k=1}^{N} T_{d_k}$ be the set of indices for all tokens belonging to any document segment. For each signal token $t_i \in T_{q,signal}$ at layer $l^*$, we compute attention scores towards document tokens $t_j \in T_{docs}$ as following: **1.** Obtain query vectors $Q_i^{(l^*)}$ for $t_i \in T_{q,signal}$ and key vectors $K_j^{(l^*)}$ for $t_j \in T_{docs}$. **2.** Compute raw attention logits $z_{ij} = Q_i^{(l^*)}(K_j^{(l^*)})^T/\sqrt{d_k}$ for all $t_j \in T_{docs}$. **3.** Compute normalized attention probabilities *only over the document tokens*: $\alpha'_{ij} = \text{softmax}_j(z_{ij})$, where the softmax is computed across all $j$ such that $t_j \in T_{docs}$. This normalization focuses the probability mass exclusively on the candidate documents, ignoring instructions and query tokens. **4.** Aggregate these probabilities to compute an attention mass score for each document $d_k$: $S(q, d_k) = \sum_{t_i \in T_{q,signal}} \sum_{t_j \in T_{d_k}} \alpha'_{ij}$ (Alternative: could use mean aggregation over $t_i$). This score $S(q, d_k)$ quantifies the relevance signal from the carrier tokens towards document $d_k$. **5.** Apply a contrastive loss using these scores. We use the InfoNCE loss with temperature $\tau$:

$$\mathcal{L}_{\text{aux}} = \mathcal{L}_{\text{InfoNCE}}(S(q, d^*), \{S(q, d_k)\}_{k \neq *}; \tau) = -\log \frac{\exp(S(q, d^*)/\tau)}{\sum_{k=1}^{N} \exp(S(q, d_k)/\tau)} \quad (2)$$

where $d^*$ is the ground-truth relevant document. This loss encourages the score $S(q, d^*)$ for the relevant document to be higher than scores for irrelevant documents.

**Overall Training Objective.** The BlockRank model is fine-tuned by minimizing a combined loss function that includes both the standard next-token prediction objective and our auxiliary attention loss:

$$\mathcal{L}_{Total} = \mathcal{L}_{NTP} + \lambda \mathcal{L}_{\text{aux}} \quad (3)$$

Here, $\mathcal{L}_{NTP}$ is the cross-entropy loss calculated on the answer tokens (similar to standard instruction tuning) based on the model's prediction of the next token in the sequence, computed using the

final hidden states which are generated respecting the structured attention masks defined in Section 4.1. $\mathcal{L}_{\text{aux}}$ is the auxiliary InfoNCE loss defined in Equation 2, applied only at layer $l^*$. $\lambda$ is a hyperparameter balancing the two losses (we use $\lambda = 0.1$ in our experiments).

### 4.3 Efficient Attention-Based Inference

An advantage of BlockRank is that the auxiliary loss explicitly optimizes the attention scores $S(q, d_k)$ to reflect relevance. This allows for an alternate efficient inference mechanism during the prefill stage of the context processing. It can bypass the iterative auto-regressive decoding process, and even the full forward pass (depending on the choice of $l^*$). The inference mechanism can be defined as follows: **1.** Given a $\text{prompt}(q, D)$, perform a partial forward pass of the BlockRank model up to the target middle layer $l^*$. **2.** Compute the document relevance scores $S(q, d_k)$ for all candidate documents $k \in \{1, \ldots, N\}$ using the exact same procedure as described for the auxiliary loss calculation (Section 4.2, steps 1-4), utilizing the signal carrier tokens $T_{q,signal}$ and performing the softmax over document tokens $T_{docs}$ only. **3.** Identify the index $\hat{k}$ of the document with the highest score: $\hat{k} = \arg\max_k S(q, d_k)$. **4.** Output the corresponding document identifier $id_{\hat{k}}$, for top-$K$ predictions output $\arg\text{top}_k S(q, d_k)$

## 5 Experimental Results

This section empirically evaluates the proposed BlockRank method. We conduct two sets of experiments: first, an evaluation on the BEIR benchmark to assess zero-shot generalization against state-of-the-art re-rankers, and second, a controlled in-domain evaluation to analyze effectiveness, efficiency, and scalability. We aim to answer the following research questions: (**RQ1**) How does BlockRank compare against strong baselines in terms of retrieval effectiveness, both in zero-shot generalization and in-domain settings? (**RQ2**) What are the efficiency benefits of BlockRank compared to standard fine-tuning, particularly when scaling the number of in-context documents? (**RQ3**) What is the contribution of BlockRank's core components (structured sparse attention, auxiliary attention loss, and attention-based inference) to its overall performance?

### 5.1 Experimental Setup

**Goal & Task.** Given a query and a list of candidate documents retrieved by an initial, potentially weaker retriever, the goal is to identify the most relevant document(s) from *within that list* by processing the entire list in the LLM's context.

**Datasets & Formatting.** For assessing zero-shot generalization, we use 11 diverse datasets from the BEIR benchmark [Thakur et al., 2021] replicating Table 1 in Reddy et al. [2024]. In this setting, the task is to rerank the top-100 documents provided by Contriever [Izacard et al., 2021] model. For in-domain analysis, we use two standard passage retrieval benchmarks: MSMarco Passage Ranking [Bajaj et al., 2018] and Natural Questions (NQ) [Kwiatkowski et al., 2019]. During training, we construct candidate lists for each query by retrieving an initial set of 30 passages using a pre-trained sentence transformer model with teacher-forcing (i.e. always adding ground-truth documents). This list is then formatted into the prompt structure shown in Figure 3. During in-domain evaluation, we construct lists of varying sizes ($N = 10$ to $500$) to test scalability. More details can be found in Appendix B.

**Evaluation.** We evaluate model performance on two primary aspects: effectiveness and efficiency. For BEIR, effectiveness is measured using nDCG@10. For in-domain experiments on MSMarco and NQ, we report Precision@1 and Mean Reciprocal Rank @ 10 (MRR@10). Efficiency is quantified by Inference Latency, the end-to-end wall-clock time per query.

**Baselines.** We compare BlockRank against a comprehensive set of baselines tailored to each experimental setting. For the BEIR generalization benchmark, we compare against contemporary listwise re-rankers, including a strong cross-encoder, RankVicuna [Pradeep et al., 2023a], RankZephyr [Pradeep et al., 2023b], and the recent state-of-the-art model, FIRST [Reddy et al., 2024]. For in-domain analysis, our primary comparison is with Full Fine-tuning (Full-FT) (full causal attention with only NTP loss) of the same base model and the same training data. We also include results from zero-shot LLMs (Mistral-7B-Instruct, Gemini-2.0-flash). For broader context, we include

Table 1: nDCG@10 on BEIR benchmark, all re-ranker rank top-100 documents retrieved from Contriever retrieval model. **Bold** indicates the best numbers.

| Reranker | Train Data | Avg. | Climate-FEVER | DB-Pedia | FEVER | FiQA | Hotpot QA | MS Marco | NF-Corpus | NQ | Sci-docs | Sci-fact | Trec-COVID |
|---|---|---|---|---|---|---|---|---|---|---|---|---|---|
| None (Contriever) | MS Marco | 45.9 | 23.7 | 41.3 | 75.8 | 32.9 | 63.8 | 40.7 | 32.8 | 49.8 | 16.5 | 67.7 | 59.6 |
| Cross-Encoder | MS Marco | 50.7 | 25.5 | 47.0 | 81.9 | 35.6 | 71.8 | 47.0 | 34.5 | 57.6 | 17.0 | 69.1 | 71.0 |
| Rank Vicuna | GPT 3.5 | 50.7 | **28.2** | 50.0 | 81.0 | 35.9 | 73.5 | 36.7 | 33.1 | 58.6 | 18.4 | 70.5 | 71.3 |
| Rank Zephyr | GPT 3.5+4 | 53.7 | 25.6 | 50.0 | 80.1 | 42.2 | 71.6 | 42.7 | **37.7** | 65.6 | **20.5** | **76.7** | 78.4 |
| FIRST | GPT-4 | 54.3 | 26.7 | **50.9** | 81.7 | 42.2 | 74.2 | 44.4 | 37.4 | **66.4** | 20.4 | 74.6 | **78.8** |
| BlockRank Mistral | MS Marco | **54.8** | 26.8 | 49.7 | **87.3** | **44.9** | 75.5 | **48.6** | 36.6 | 62.4 | 18.7 | 76.5 | 76.2 |

Traditional Retrieval Models such as the lexical baseline BM25, the dense retriever GTR [Ni et al., 2021], ColBERTv2 [Santhanam et al., 2022], and best performing Sentence Transformer Encoders specific to each dataset (`msmarco-distilbert-dot-v5` for MSMarco and `all-MiniLM-L12-v2` for NQ). Furthermore, we consider pairwise cross-encoder baselines like monoBERT [Nogueira and Cho, 2020] and improved versions of monoT5 [Nogueira et al., 2020].

**Implementation Details.** BlockRank and the Full-FT baseline utilize `Mistral-7B-v0.3` as the base model. For fine-tuning both models, we employ the Adafactor optimizer [Shazeer and Stern, 2018] with a learning rate of $3 \times 10^{-7}$ and a global batch size of 32 (accumulated across replicas). Each model is trained for 1 epoch with a linear warmup followed by cosine decay. For BlockRank, the auxiliary loss weight $\lambda$ is set to 0.1, and $\tau$ is set to 0.05. Unless stated otherwise, BlockRank results employ the proposed attention-based inference. Decoding based experiments with BlockRank, LLM baselines (Full-FT Mistral and Zero-Shot LLMs) utilize greedy decoding to generate the relevant document identifier(s); to get multiple predictions (for MRR@10 evaluation) we use constrained beam decoding with beam-size set to 10, where only valid outputs are generated. All LLM fine-tuning and inference experiments were conducted using `JAX` on Google Cloud TPUs (specifically, 8 chip `v6e` configuration), and reported efficiency metrics correspond to this setup as well.

## 5.2 Main Performance Comparison

**Generalization to Diverse Tasks (RQ1)** The results in Table 1 show that MSMarco-trained BlockRank Mistral (54.8) outperforms FIRST (54.3), RankZephyr (53.7), and RankVicuna (50.7), demonstrating strong out-of-distribution generalization. Importantly, BlockRank achieves strong results with the significant efficiency gains (Figure 4), presenting a compelling combination of effectiveness and scalability. Furthermore, it gets the strong performance by processing the entire list of 100 candidate documents in a single forward pass instead of multiple sliding-window forward passes over the candidate set – which is required for other listwise ranking models. These results also indicate that BlockRank is not sensitive to the first-stage retriever, as it effectively ranks candidates from Contriever despite its training data being constructed with a different retrieval model.

**In-Domain Performance (RQ1)** Table 2 summarizes the quality comparisons on the NQ and MSMarco in a controlled environment where both BlockRank and Full-FT Mistral are trained on the same training data and evaluated on in-domain data, for broader comparison we also provide results for additional baselines trained on the same data. Our proposed BlockRank consistently outperforms its direct counterpart, Full-FT Mistral (7B).

**Scalability (RQ2).** Figure 4 underscores BlockRank's substantial inference efficiency advantage over the Full-FT baseline as the number of in-context documents ($N$) increases. BlockRank model consistently exhibits lower latency; at $N = 100$, it is approximately $4.7\times$ faster. More critically, its latency scales linearly with $N$, reaching 1.15s at $N = 500$. Furthermore, BlockRank model maintains its P@1 (peaking around 29.2% for $N = 200$ and remaining at 28.7% for $N = 500$), whereas Full-FT's P@1 sharply degrades beyond $N = 100$ (dropping to $\approx 26.7\%$ at $N = 500$).

Figure 4: P@1 and Latency (annotated) of BlockRank vs Full-FT Mistral, scaling $N$ on MSMarco.

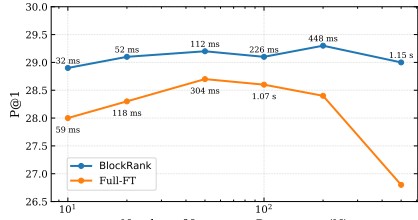

Table 2: Comparison on MSMarco and NQ datasets in controlled settings. Encoder methods are evaluated on the full corpus while the rest of the baselines are evaluated on a shortlist. Best results are highlighted in **Bold**.

| Category | Method | Model Size | NQ | MSMarco | |
|---|---|---|---|---|---|
| | | | Precision@1 | Precision@1 | MRR@10 |
| Sparse Retrieval | BM25 | - | 29.7 | - | 18.4 |
| Dual-Encoder | Sentence-transformer [Reimers and Gurevych, 2019] | 66M | 58.8 | 24.8 | 37.2 |
| | GTR-XXL [Ni et al., 2021] | 4.8B | - | - | 38.8 |
| | ColBERTv2 [Santhanam et al., 2022] | 110M | - | - | 39.7 |
| Cross-Encoder | monoBERT [Nogueira and Cho, 2020] | 110M | - | - | 38.2 |
| | monoT5-XL [Nogueira et al., 2020] | 3B | - | - | 41.2 |
| Zero-Shot LLM | Mistral-7B-v0.3-it [Jiang et al., 2023] | 7B | 43.5 | 13.1 | - |
| | Gemini-2.0-flash [Team et al., 2023] | - | 65.1 | 16.9 | - |
| Fine-tuned LLM | Full-FT Mistral | 7B | 75.5 | 28.7 | 38.3 |
| | **BlockRank Mistral (Ours)** | 7B | **76.2** | **29.1** | **42.0** |

## 5.3 Ablation Studies

To understand contribution of components of BlockRank (RQ3), we perform several ablation experiments, primarily on the MSMarco dataset with $N = 50$. More ablation is provided in Section D of Appendix.

**Impact of Training Loss** Table 3 ablates the contributions of $\mathcal{L}_{NTP}$ and $\mathcal{L}_{aux}$ to P@1, evaluated with both auto-regressive and attention-based inference. Introducing $\mathcal{L}_{aux}$ consistently enhances performance for attention-based inference, and for BlockRank (which incorporates structured attention), it increases from 27.8 to 29.1. As expected, the $\mathcal{L}_{NTP}$ objective is crucial for generative decoding performance, as seen by the sharp drop in Decode Prec@1 for 'BlockRank (w/o ntp)' to 15.8. Notably, the full BlockRank configuration achieves the highest Attn Prec@1 (29.1), demonstrating that $\mathcal{L}_{aux}$ effectively optimizes attention scores for direct retrieval, making attention-based inference the preferred mode for our method.

Table 3: Impact of training loss on Attention-based (Attn) and Decoding (Decode) Inference.

| Training Configuration | Precision@1 | |
|---|---|---|
| | Decode | Attn |
| Full-FT | 28.7 | 27.6 |
| Full-FT (w/ aux) | 28.7 | 28.1 |
| BlockRank (w/o ntp) | 15.8 | 28.6 |
| BlockRank (w/o aux) | 28.4 | 27.8 |
| **BlockRank (full)** | **28.7** | **29.1** |

**Impact of Inference Method** Table 4 ablates the inference, comparing decoding against our attention-based approach on P@1 and MRR@10; for Full-FT, Decode MRR@10 uses a beam size of 10. The results show that while auto-regressive decoding yields comparable P@1 for both Full-FT (28.7) and BlockRank (28.7) models, it is significantly less effective at producing a strong ranked list for MRR@10. In contrast, BlockRank with attention-based inference performs best, achieving a notably better MRR@10 (42.0). BlockRank's attention-based inference, optimized via its auxiliary loss, is more calibrated at assigning relevance across multiple predictions.

Table 4: Ablation: Inference Method Effectiveness & Latency (MSMarco, N=50).

| Model | Inference Method | P@1 | MRR@10 |
|---|---|---|---|
| Full-FT | Decode | 28.7 | 38.4 |
| Full-FT | Attn | 27.6 | 38.8 |
| BlockRank | Decode | 28.7 | 40.0 |
| **BlockRank** | **Attn** | **29.1** | **42.0** |

## 6 Conclusion

This work addresses the efficiency challenge in In-Context Retrieval (ICR) by analyzing LLM attention, identifying structured sparsity and query-token retrieval signals. We introduced BlockRank, a method that enforces this task-specific sparsity for linear complexity and uses a contrastive auxiliary loss to directly optimize these internal attention signals for relevance. Experiments on MSMarco and NQ show BlockRank (Mistral-7B) matches or surpasses standard fine-tuning effectiveness while being significantly more efficient at inference and training. This offers a scalable and effective approach for LLM-based ICR. However, we acknowledge our current findings are primarily demonstrated on a specific model architecture, and the robustness of the learned attention signals for direct inference across highly diverse tasks needs more investigation.

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

# A  Societal Impact

The BlockRank methodology, by enhancing the efficiency and scalability of In-context Retrieval (ICR) in Large Language Models (LLMs), makes advanced semantic retrieval more computationally tractable and can democratize access to powerful information discovery tools. This could accelerate research, improve educational outcomes by providing more relevant information quickly, and empower individuals and organizations with better decision-making capabilities. Furthermore, the increased efficiency directly translates to reduced energy consumption for retrieval-intensive LLM applications, contributing to more environmentally sustainable AI development and deployment. By enabling effective ICR on potentially smaller or more optimized models, BlockRank could also broaden the reach of these technologies in resource-constrained environments.

However, like many advancements in AI, more efficient information retrieval also presents challenges. The underlying LLMs can inherit and potentially amplify societal biases present in their training data. Therefore, continued research in this area should be accompanied by a strong emphasis on transparency, and the development of robust mechanisms to identify and mitigate the spread of harmful or misleading content.

# B  Dataset and Hyperparameter Details

This section provides a detailed description of the dataset and hyperparameters used in this study to ensure reproducibility.

## B.1  Datasets

We use two standard passage retrieval benchmarks:

- **MSMarco Passage Ranking** [Bajaj et al., 2018]: We use MSMarco v1 passage retrieval dataset, it has total $8.8M$ passages, $\sim 500K$ training queries and $6980$ validation queries. We directly utilize the hard negatives collection[1] from huggingface for training. During test we retrieve the top-$N$ passages using `msmarco-distilbert-dot-v5` sentence-transformer.
- **Natural Questions (NQ320K)** [Kwiatkowski et al., 2019]: We use NQ320K passage retrieval dataset which has $\sim 320K$ passages in the corpus, $\sim 300K$ training queries and $7830$ validation queries. For NQ, we collect hard negatives using `all-MiniLM-L12-v2` sentence-transformer model for training. We use the same model during inference as well to retrieve top-$N$ passages.

## B.2  Fine-tuning Details (**BlockRank** and Full-FT)

The following fine-tuning settings were used for both BlockRank and Full-FT Mistral-7B:

- **Optimizer**: Adafactor [Shazeer and Stern, 2018] with $\beta_1 = 0.9$.
- **Learning Rate**: $3 \times 10^{-7}$.
- **Learning Rate Schedule**: Linear warmup for 50 steps followed by a cosine decay.
- **Batch Size**: A global batch size of 32, accumulated across replicas.
- **Number of Epochs**: 1 epoch.
- **Weight Decay**: No weight decay.
- **Gradient Clipping**: gradient norm clipped to 1.0.
- **Loss for Full-FT**: Standard Next Token Prediction (NTP) cross-entropy loss, calculated on the answer tokens (i.e., the ID of the relevant document).

---

[1]https://huggingface.co/datasets/sentence-transformers/msmarco-msmarco-distilbert-base-v3

### B.3 BlockRank Specific Hyperparameters

In addition to the general fine-tuning settings, the following hyperparameters are specific to the BlockRank method:

- **Auxiliary Loss Weight** ($\lambda$): The hyperparameter balancing the NTP loss and the auxiliary attention loss ($\mathcal{L}_{aux}$) was set to $\lambda = 0.1$ this ensures that both loss have the same scale.
- **InfoNCE Temperature** ($\tau$): The temperature parameter for the InfoNCE loss ($\mathcal{L}_{aux}$) was set to $\tau = 0.05$.
- **Signal-Carrying Query Tokens** ($T_{q,signal}$): Based on our prompt template and empirical analysis (Section D), the set of tokens for signal-carrying query tokens was $T_{q,signal} = [":", "['"]$.
- **Middle Layer for Auxiliary Loss** ($l^*$): The auxiliary loss $\mathcal{L}_{aux}$ was applied at a specific middle layer $l^* = 20$, determined empirically as described in Section D.3.
- **Chunk Length** ($L_{chunk}$): The fixed length for chunks used in the structured attention mechanism. We set $L_{chunk} = 160$ for MSMarco and $L_{chunk} = 384$ for NQ, this ensures that $\sim 95\%$ of the passages get full represented in $L_{chunk}$ sequence length.

## C   Attention Complexity Analysis

This section provides a analysis of the computational complexity of the structured attention mechanism within a single layer of the BlockRank model architecture. Our aim is to clearly illustrate the scalability benefits of BlockRank, particularly its linear scaling with respect to the number of candidate documents, $N$. We define $L_{chunk}$ as the fixed characteristic length (number of tokens) for segments after processing, and $d$ as the hidden dimension of the model. For this analysis, we assume that the instruction segment, each of the $N$ document segments, and the query segment have effective lengths $L_{Inst} = L_{chunk}$, $L_{doc} = L_{chunk}$, and $L_q = L_{chunk}$ respectively, when their attention computations are considered. This section focuses exclusively on the attention component's complexity, as this is where BlockRank introduces its primary architectural modification for efficiency.

The BlockRank model implements a structured sparse attention mechanism, as detailed in Section 4.1 of the main paper, where different parts of the input prompt adhere to distinct attention patterns. The instruction segment, with its effective length of $L_{chunk}$, performs causal self-attention, leading to a complexity of $C_{attn,Inst} = \mathcal{O}(L_{chunk}^2 \cdot d)$. For the $N$ document segments, each also of effective length $L_{chunk}$, tokens attend both within their own segment and to tokens within the instruction segment. This means the effective context length for a token in any given document segment is $L_{chunk} + L_{chunk} = 2L_{chunk}$. Consequently, the attention complexity for a single document segment is $\mathcal{O}(L_{chunk} \cdot 2L_{chunk} \cdot d) = \mathcal{O}(2L_{chunk}^2 \cdot d)$. Summing across all $N$ document segments, their total attention complexity is $C_{attn,Doc} = N \cdot \mathcal{O}(2L_{chunk}^2 \cdot d)$.

The query segment, also with an effective length of $L_{chunk}$, has the full attention scope. It attends to its own tokens, tokens from the instruction segment, and tokens from all $N$ document segments. The total context length for these query tokens becomes $L_{chunk} + L_{chunk} + (N \cdot L_{chunk}) = (N + 2)L_{chunk}$. The attention complexity for the query segment is therefore $C_{attn,Query} = \mathcal{O}(L_{chunk} \cdot (N + 2)L_{chunk} \cdot d) = \mathcal{O}((N + 2)L_{chunk}^2 \cdot d)$.

Summing the complexities of these components gives the total attention complexity per layer for the BlockRank model, $C_{attn,\text{BlockRank}}$:

$$C_{attn,\text{BlockRank}} = C_{attn,Inst} + C_{attn,Doc} + C_{attn,Query}$$

$$C_{attn,\text{BlockRank}} = \mathcal{O}(L_{chunk}^2 d) + N \cdot \mathcal{O}(2L_{chunk}^2 d) + \mathcal{O}((N + 2)L_{chunk}^2 d)$$

This simplifies to $\mathcal{O}(3L_{chunk}^2 d + 3NL_{chunk}^2 d)$, which is $\mathcal{O}((N + 1)L_{chunk}^2 d)$. The dominant term thus yields a total attention complexity of $C_{attn,\text{BlockRank}} = \mathcal{O}(N \cdot L_{chunk}^2 \cdot d)$. This result clearly shows that the attention complexity in the BlockRank architecture scales linearly with $N$, the number of documents.

In contrast, a standard Transformer model processing a sequence of comparable total length $S \approx (N + 2)L_{chunk}$ would exhibit an attention complexity of $C_{attn,Std} = \mathcal{O}(S^2 \cdot d)$. For large $N$, this is approximately $\mathcal{O}(((N + 2)L_{chunk})^2 \cdot d) = \mathcal{O}(N^2 \cdot L_{chunk}^2 \cdot d)$, which is quadratic with respect to $N$.

# D Additional Results

## D.1 Calibration Problem in Beam Decoding with Full-FT Model

To analyze the behavior of the standard fine-tuned (Full-FT) model when generating multiple distinct predictions via beam decoding, we conducted an entropy analysis on the individual tokens of the predicted document identifiers. This experiment was designed to assess the diversity of predictions for structured identifiers (two-digit IDs from 0-99, given $N = 100$ candidate documents). For each query in the test set, we generated 10 unique document ID predictions from the Full-FT model. We then computed the entropy of the distribution of the first digit ($id_0$) and the second digit ($id_1$) across these 10 predictions. This was compared against the

Table 5: Entropy of predicted document ID digits ($id_0$ and $id_1$) over 10 beam-decoded predictions for the Full-FT model versus random predictions. Lower entropy indicates less diversity in the generated digits across the prediction list.

| Prediction Model | Entropy $id_0$ | Entropy $id_1$ |
|---|---|---|
| Full-FT | $2.28 \pm 0.43$ | $2.19 \pm 0.46$ |
| BlockRank | $2.54 \pm 0.24$ | $2.67 \pm 0.24$ |
| Random | $2.55 \pm 0.25$ | $2.66 \pm 0.24$ |

entropy derived from 10 randomly drawn unique two-digit IDs. Because the candidate list is randomly shuffled and the ID assigned to each document is completely independent from its content, a lower entropy would indicate a undesirable concentration of predicted digits, suggesting a lack of diversity in the generated list beyond the top few candidates.

The results, summarized in Table 5, show that the Full-FT model exhibits lower average entropy for both $id_0$ ($2.28 \pm 0.43$) and $id_1$ ($2.19 \pm 0.46$) compared to the random baseline ($2.55 \pm 0.25$ for $id_0$ and $2.66 \pm 0.24$ for $id_1$). This decreased entropy indicates that the sequence of document identifiers generated by the Full-FT model via beam decoding tends to be less diverse in its constituent digits than random chance would suggest. To give an example, we observe that let's say the model predicts 73 as it's top prediction with high confidence, then there is a high likelihood that it will predict other IDs either starting with 7 or ending with 3. Such concentration implies that while the model may identify a strong top candidate, its ability to produce a well-calibrated and varied set of subsequent predictions is limited, due to the nature of auto-regressive log-probability distributions. This observation supports the main paper's discussion (Section 5.3, Table 4) on the sub-optimality of beam decoding for generating ranked lists for ICR.

## D.2 Analysis of Retrieval Signals in Attention Patterns of Full-FT Mistral

To substantiate the claims made in Section 3 of the main paper regarding the presence of retrieval signals within the internal attention patterns of a standard fine-tuned (Full-FT) language model, we conducted a series of analytical experiments. These experiments, detailed below, confirm the characteristics of such signals using attention-based inference on the MSMarco dev dataset.

First, we investigate the specific carriers of the retrieval signals. Figure 5 presents the P@1 performance of attention-based inference when attention scores are extracted from different query tokens within the prompt. This analysis reveals that certain query tokens, particularly those located towards the end of the query or specific delimiter tokens such as ":" and terminal prompt markers, serve as strong "signal carriers," yielding significantly higher P@1 when their attention patterns are used to predict the relevant document.

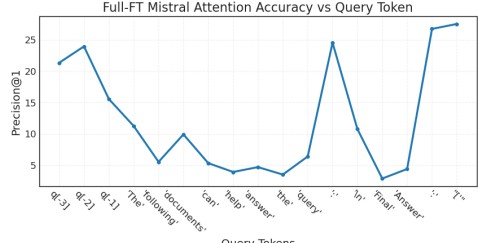

Figure 5: Performance of Full-FT model's attention-based inference vs the query token for which attention scores are extracted from.

Complementing this, Figure 6 evaluates both P@1 and Mean Reciprocal Rank @10 for attention-based inference as a function of the Transformer layer index from which attention scores are derived. This experiment confirms that the retrieval signal is most prevalent in the middle layers of the Full-FT model, with performance declining in earlier and later layers. Collectively, these empirical findings

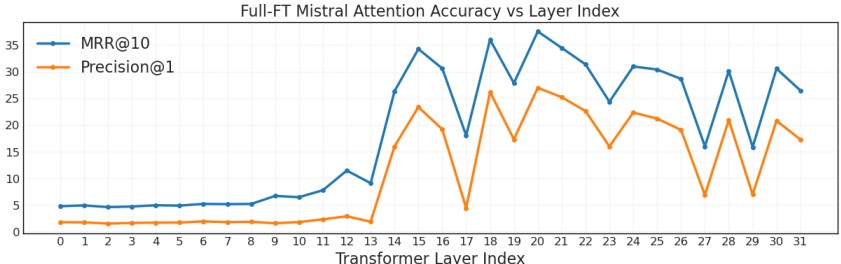

Figure 6: Performance of Full-FT model's attention-based inference as a function of the Transformer Layer Index from which attention scores are extracted (MSMarco).

demonstrate that standard LLMs fine-tuned for In-context Retrieval exhibits latent retrieval signals within their attention mechanisms. These signals are characterized by their preferential emergence in middle layers and their association with specific query tokens.

### D.3 Layerwise Emergence of Retrieval Signals and Choice of $l^*$

Figure 7 illustrates the evolution of layerwise Precision@1 derived from attention scores on a held-out subset of MSMarco training data as the Full-FT model undergoes training. It is observed that effective retrieval signals, as measured by the attention-P@1 metric, do not develop uniformly across all layers. Instead, they emerge more prominently and strengthen considerably in the middle layers of the transformer (layers 12 through 24) as training progresses, while shallower and deeper layers exhibit comparatively weaker signal strength. Based on this we set the $l^* = 20$ for all of our BlockRank experiments. Although, we find that the choice of $l^*$ in BlockRank is not very sensitive to this specific layer, any reasonable middle layer gives similar performance.

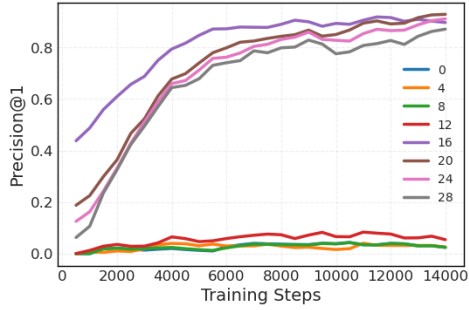

Figure 7: Layerwise Attention Precision@1 on a held-out subset of MSMarco training data vs training steps for Full-FT model

### D.4 Impact of Including Query in Prompt Prefix

We investigated whether providing the query context upfront, in addition to its standard position at the end of the prompt, impacts retrieval performance. Table 6 compares the Precision@1 results on MSMarco for the Full-FT baseline and our BlockRank model using the standard format (query only at end; denoted by ✗ in the table) versus the query-prefix format (query at beginning and end; denoted by ✓). Including the query redundantly in the prefix (✓) improved performance over the standard format (✗) for

Table 6: Ablation on Including Query in Prompt Prefix. Comparison of Precision@1 on MSMarco (N=100) for Full-FT and BlockRank models.

| Model | Query in Prefix | Prec@1 |
|---|---|---|
| Full-FT-Mistral | ✗ | 27.2 |
| Full-FT-Mistral | ✓ | 28.7 |
| BlockRank-Mistral | ✗ | 24.2 |
| **BlockRank-Mistral** | ✓ | 29.1 |

both models. The Full-FT model's Prec@1 increased from 27.2 to 28.7 (+1.5), while our BlockRank model saw a more substantial increase from 24.2 to 28.1 (+3.9). This suggests that priming the model with the query context before it processes the candidate documents is beneficial, perhaps allowing attention mechanisms, particularly the specialized ones in BlockRank, to focus more effectively on query-relevant information throughout the sequence. Given this clear advantage, we utilize the prompt format that includes the query in the prefix for all other reported experiments.

## D.5 Cross-dataset Generalization

To assess the generalization of the BlockRank models, we evaluated BlockRank Mistral models trained on one dataset and tested on another, unseen dataset. Specifically, models were fine-tuned separately on the MSMarco and Natural Questions (NQ) training sets, and their Precision@1 (P@1) performance was subsequently measured on the test sets of both NQ and MS-Marco. For reference, we also include the performance of a zero-shot Mistral-7B-instruct model (denoted as *No Training*). The results of this cross-dataset evaluation are presented in Table 7. As expected, BlockRank models achieve their best performance when evaluated on the in-domain test set. When evaluated on out-of-domain datasets, the performance, shows positive transfer above the *No Training* baseline but is considerably lower than in-domain scores.

Table 7: Cross-dataset generalization performance of BlockRank Mistral models. P@1 scores are reported on the NQ and MSMarco test sets for models with no training (zero-shot Mistral-7B-instruct), fine-tuned on NQ, and fine-tuned on MSMarco.

| Training Data | NQ P@1 | MSMarco P@1 |
|---|---|---|
| *No Training* | 43.5 | 13.1 |
| NQ | 76.2 | 18.2 |
| MSMarco | 62.0 | 29.1 |

