# OpenReview forum: "Scalable In-context Ranking with Generative Models"
_NeurIPS.cc/2025/Conference — NeurIPS 2025 poster_

### Official Review · Reviewer_iUgW · 2025-06-24

**Clarity:** 3
**Significance:** 3
**Originality:** 3
**Rating:** 4
**Confidence:** 3

**Summary:**

This paper presents ReTuning, a fine-tuning method for improving the efficiency and effectiveness of large language models (LLMs) on In-context Retrieval (ICR) tasks. The authors identify structured attention patterns in LLMs fine-tuned for retrieval and propose two key innovations: (1) a structured sparse attention mechanism that reduces inference complexity, and (2) an auxiliary contrastive loss that explicitly aligns query-token attention with relevant documents. The proposed method achieves competitive retrieval accuracy while being significantly faster than standard fine-tuning baselines on benchmarks such as MS MARCO and Natural Questions.

**Questions:**

See weaknesses.

**Ethical Concerns:**

["NO or VERY MINOR ethics concerns only"]

**Final Justification:**

The response addresses most of my concerns, and I keep my positive assessment.

**Limitations:**

yes

**Paper Formatting Concerns:**

No paper formatting concerns.

**Quality:**

3

**Strengths And Weaknesses:**

**Strengths:**
1. The paper provides a clear empirical analysis of attention patterns in fine-tuned LLMs, revealing structured sparsity and signal-carrying query tokens.

2. The proposed attention-based inference mechanism avoids auto-regressive decoding. This idea is interesting.

3. ReTuning outperforms or matches strong baselines on ICR tasks with notable improvements in inference efficiency, especially as the number of candidate documents scales up.

**Weaknesses:**
1. The structured sparse attention design is similar to block-wise attention patterns used in prior works such as "Block Attention for Efficient Prefilling", which the paper does not acknowledge, potentially overclaiming novelty.

2. The method relies on pre-defined signal tokens (e.g., “:”, “[”) for attention supervision, but it remains unclear how the system handles general queries without such tokens.

3. The attention-based scoring mechanism depends on extracting attention weights from intermediate layers, making it incompatible with techniques like FlashAttention, which could significantly hinder scalability in long-context scenarios.

4. The use of raw summed attention over document tokens may introduce length bias, favoring longer documents regardless of relevance.

---

> ### Author Rebuttal · Authors · 2025-07-31
>
> We sincerely thank the reviewer for their feedback and questions. We are glad they found our analysis and motivation clear. We provide clarifications below that we hope will address the concerns and solidify the reviewer's positive assessment.
>
> ---
>
> ## Novelty vs. Block Attention (Weakness 1)
> We would like to note that while 'block attention' and other block-wise sparse methods are known techniques for general efficiency in LLMs, our work's novelty lies in its *task-specific design and integration into a complete training and inference system for In-context Retrieval*.
>
> Moreover, we provide contributions beyond structured attention:
> - **Analysis of Emergent Retrieval Signals**: Our paper provides a valuable analysis (Section 3 and Appendix D.2, D.3, D.4) of how LLMs develop latent "retrieval heads" for the ICR task. This analysis itself is a contribution that augments the emerging literature on understanding the internal mechanisms of LLMs.
> - **Training Objective**: this analysis directly informs the design of our auxiliary contrastive loss ($L_{aux}$) and alternate retrieval inference mechanisms. To the best of our knowledge, this is a novel technique for directly optimizing and calibrating the internal self-attention scores of an LLM for retrieval.
>
> ---
> ## Robustness and Technical Limitations (Weaknesses 2, 3, 4)
>
> **(Weakness 2) Generalizability of Signal Tokens**
>
> The specific signal-carrying tokens, such as ":" and "[", are not required to be present in the raw user query. Instead, they are a deliberate part of the prompt template that structures the input for the model, as shown in Figure 1. By explicitly including these tokens in the prompt template, we provide a consistent and reliable structural "anchor" for the model to attach its retrieval signal to during fine-tuning. This ensures that such signal carriers are always present for the model to utilize, regardless of the specific content of the user's query.
>
> **(Weakness 3) Incompatibility with FlashAttention**
>
> We thank the reviewer for this subtle but important practical point. Please find below our response:
> - **Fundamental Complexity Advantage**: ReTuning's primary efficiency gain comes from changing the attention complexity from quadratic, $O(N^2)$, to linear, $O(N)$. While FlashAttention optimizes the quadratic operation, it doesn't change its underlying complexity, making our method fundamentally more scalable for very long contexts.
> - **Potential for Custom Kernels**: Our attention-based inference does not require the entire attention matrix, it only needs the block-summed probabilities for a few specific query tokens. This computational pattern is highly amenable to a custom GPU kernel—similar in spirit to FlashAttention—that could compute these specific scores directly without materializing the intermediate matrix. Developing such a kernel is a promising direction for future engineering work to gain even further speedups.
> - **Flexible and Compatible Inference**: Our method offers two inference pathways. While our attention-based inference is incompatible with standard FlashAttention, our standard decoding path is fully compatible with it and achieves similar performance to the baseline (as shown in Table 2).
>
> **(Weakness 4) Potential for Length Bias**
>
> We acknowledge that summing attention scores could potentially introduce length bias. Our framework, however, can easily mitigate this. We primarily used sum aggregation in our experiments and found it performed well on our benchmarks, which have natural passage length variations. However, as noted in the paper (line 233), our relevance score calculation can be trivially modified to use mean aggregation over a document's tokens instead of a sum. This provides explicit length normalization. We will expand on this point in the final version and add an ablation study comparing mean vs. sum aggregation to demonstrate this robustness.
>
>
> ---
>
> We hope our response has addressed the reviewer's concerns. We thank them again for their valuable feedback and are happy to provide any further clarifications/results during the discussion period.

---

> > ### Comment · Reviewer_iUgW · 2025-08-05
> >
> > Thank you for the clarification. While some of my concerns have been addressed, the discussion regarding FlashAttention compatibility remains insufficiently detailed. Specifically, it would be important to provide concrete implementation details on how individual documents are encoded separately, including how documents of varying lengths are handled during batching. These factors can have a substantial impact on actual runtime performance. To substantiate the claimed scalability benefits, empirical efficiency metrics such as TTFT or throughput under long-context scenarios should also be provided.

---

> > > ### Author Response · Authors · 2025-08-05
> > > **Response to follow-up questions (part 1)**
> > >
> > > We thank the reviewer for the follow-up questions. We believe the implementation details are present in the paper, but we acknowledge that they are spread across several sections (Sec 4.1, Sec 5, Appendix B, Appendix C).
> > >
> > > To consolidate and clarify, we first provide a simplified, JAX-like code of our method. We will then explain how this can map to an I/O-aware, FlashAttention-style implementation and conclude by answering the specific questions with empirical metrics.
> > >
> > > ## Simplified JAX code of a ReTuning layer and Relevance computation
> > > ```python
> > > import jax
> > > import jax.numpy as jnp
> > > from flax import linen as nn
> > > from einops import rearrange
> > >
> > > # Shapes: N=docs, L=chunk_length, D=hidden_dim, H=num_heads, M=num_signal_tokens
> > >
> > > def standard_attn(q, k, v):
> > >     """Standard attention computation"""
> > >     logits = jnp.einsum('qhd,khd->hqk', q, k) / jnp.sqrt(q.shape[-1])
> > >     probs = nn.softmax(logits, axis=-1)
> > >     return jnp.einsum('hqk,khd->qhd', probs, v)
> > >
> > > def compute_attn_rel_probs(q, k_docs):
> > >     """Compute attention relevance probabilities from signal query tokens to docs"""
> > >     logits = jnp.einsum('mhd,nlhd->mnlh', q, k_docs) / jnp.sqrt(q.shape[-1])
> > >     probs = nn.softmax(rearrange(logits, 'm n l h -> m h (n l)'), axis=-1)
> > >     doc_probs = jnp.mean(rearrange(probs, 'm h (n l) -> m h n l', n=k_docs.shape[0]), axis=(-1, 0, 1)) # (N,)
> > >     return doc_probs # (N,)
> > >
> > > def retuning_attention_layer(
> > >     inst_h: jax.Array,   # (L, D)
> > >     docs_h: jax.Array,   # (N, L, D)
> > >     query_h: jax.Array,  # (L, D)
> > >     params: dict,        # W_q, W_k, W_v, W_o weights
> > >     signal_indices: jax.Array = None # Optional, for relevance scores
> > > ) -> tuple[jax.Array, jax.Array, jax.Array | None]:
> > >     """Computes one layer of ReTuning's structured sparse attention and optionally relevance scores."""
> > >     # 1. Project inputs to Q, K, V and reshape for multi-head attention
> > >     q_inst, k_inst, v_inst = [rearrange(inst_h @ params[w], 'l (h d) -> l h d', h=H) for w in ('W_q', 'W_k', 'W_v')]
> > >     q_docs, k_docs, v_docs = [rearrange(docs_h @ params[w], 'n l (h d) -> n l h d', h=H) for w in ('W_q', 'W_k', 'W_v')]
> > >     q_query, k_query, v_query = [rearrange(query_h @ params[w], 'l (h d) -> l h d', h=H) for w in ('W_q', 'W_k', 'W_v')]
> > >
> > >     # 2. Instruction Attention
> > >     updated_inst_h = standard_attn(q_inst, k_inst, v_inst)  # (L, H, d_head)
> > >
> > >     # 3. Document Attention Function (attends to instruction and self)
> > >     def doc_attn_fn(q_doc, k_doc, v_doc):
> > >         k_context = jnp.concat([k_inst, k_doc], axis=0) # (2*L, H, d_head)
> > >         v_context = jnp.concat([v_inst, v_doc], axis=0) # (2*L, H, d_head)
> > >         return standard_attn(q_doc, k_context, v_context)
> > >
> > >     updated_docs_h = jax.vmap(doc_attn_fn)(q_docs, k_docs, v_docs) # (N, L, H, d_head)
> > >
> > >     # 4. Query Attention (attends to all context)
> > >     k_all_docs = rearrange(k_docs, 'n l h d -> (n l) h d')
> > >     v_all_docs = rearrange(v_docs, 'n l h d -> (n l) h d')
> > >     k_query_context = jnp.concat([k_inst, k_all_docs, k_query], axis=0)
> > >     v_query_context = jnp.concat([v_inst, v_all_docs, v_query], axis=0)
> > >
> > >     updated_query_h = standard_attn(q_query, k_query_context, v_query_context) # (L, H, d_head)
> > >
> > >     # 5. Project outputs back to hidden dimension D
> > >     updated_inst_h = rearrange(updated_inst_h, 'l h d -> l (h d)') @ params['W_o']
> > >     updated_docs_h = rearrange(updated_docs_h, 'n l h d -> n l (h d)') @ params['W_o']
> > >     updated_query_h = rearrange(updated_query_h, 'l h d -> l (h d)') @ params['W_o']
> > >
> > >     # 6. Optional calculate relevance score
> > >     relevance_scores = None
> > >     if signal_indices is not None:
> > >         relevance_scores = compute_attn_rel_probs(q_query[signal_indices], k_docs)  # (N, )
> > >
> > >     return updated_inst_h, updated_docs_h, updated_query_h, relevance_scores
> > > ```
> > >
> > > ## Potential Kernelized Implementation
> > > The functions standard_attn and compute_attn_rel_probs can be mapped to I/O-aware kernels, similar in spirit to FlashAttention, to achieve maximum efficiency.
> > >
> > > 1. Kernelizing `standard_attn`:  A standard FlashAttention kernel can be directly applied here. By processing keys and values in tiles from HBM to SRAM and fusing the softmax calculation, it can compute the final attention output without materializing the full attention matrix.
> > >
> > > 2. Kernelizing `compute_attn_rel_probs`: This function is even more amenable to a custom fused kernel. Its goal is not to compute an output vector, but an aggregated probability score. A specialized kernel would:
> > >     - Load the small `q_query[signal_indices]` into fast SRAM.
> > >     - Iterate through the `k_docs` matrix in tiles.
> > >     - For each tile, use the online softmax trick to maintain the running normalization factor across all document tiles without storing the full `(M, N*L)` logit matrix.
> > >     - Accumulate the probability numerators for each document group.
> > >     - A final normalization step after iterating through all tiles would yield the final scores.
> > >
> > > One can also fuse both of these operations to avoid redundancy of computing `q_query[signal_indices] @ k_docs`

---

> > > > ### Author Response · Authors · 2025-08-05
> > > > **Response to follow-up questions (part 2)**
> > > >
> > > > ## Answers to Specific Questions
> > > > ### Q1: How are individual documents encoded separately, including how documents of varying lengths are handled during batching?
> > > >
> > > > As detailed in Section 4.1 and Appendix B:
> > > >
> > > > - The input prompt is first logically segmented into its components (1 instruction, N documents, 1 query). To handle variable lengths, each segment is padded or truncated to a fixed sequence length, $L_{chunk}$. We chose $L_{chunk}$ to cover ~95% of document lengths in our datasets (e.g., 160 for MSMarco, and 384 for NQ).
> > > > - As shown in the `jax.vmap` call in the code provided above, these N document chunks are processed in parallel. Each document computes attention over its own small, fixed-size context ($2 L_{chunk}$). This parallel structure is how documents are encoded separately and is the key to our method's linear complexity, as it avoids forming a single ($NL_{chunk} \times NL_{chunk}$) attention matrix.
> > > >
> > > > ### Q2: Empirical efficiency metrics such as TTFT or throughput under long-context scenarios?
> > > >
> > > > For our attention-based inference, the process is non-autoregressive; it performs a single forward pass to produce all relevance scores simultaneously. Therefore, the end-to-end latency we report is directly equivalent to Time To First Token (TTFT), as there is no "second" token to generate.
> > > >
> > > > The throughput, measured in queries per second (QPS), can be defined as the reciprocal of this latency. We present the latency data from Figure 4 in the requested format below.
> > > >
> > > > | # In-context Docs (N) | Context Length (Tokens) | Latency (TTFT) | Throughput (Queries/Sec) |
> > > > | :-------------------- | :---------------------- | :------------- | :----------------------- |
> > > > | 10                    | ~1,920                  | 32 ms          | ~31.3 QPS                |
> > > > | 20                    | ~3,520                  | 52 ms          | ~19.2 QPS                |
> > > > | 50                    | ~8,320                  | 112 ms         | ~8.9 QPS                 |
> > > > | 100                   | ~16,320                 | 226 ms         | ~4.4 QPS                 |
> > > > | 200                   | ~32,320                 | 448 ms         | ~2.2 QPS                 |
> > > > | 500                   | ~80,320                 | 1.15 s         | ~0.87 QPS                |
> > > >
> > > > As the table shows, our method's latency (TTFT) and throughput scale gracefully, demonstrating its practical efficiency in long-context scenarios.
> > > >
> > > > ---
> > > >
> > > > We hope these clarifications address your remaining concerns. We are happy to provide further answers. We will also integrate this discussion into the final version of the paper to improve its clarity. Thank you again for your valuable and constructive feedback.

---

> > > > > ### Comment · Reviewer_iUgW · 2025-08-06
> > > > >
> > > > > Thank you for the author's clarification, which has basically answered my questions and is consistent with what I thought. But in real-world scenarios, it is unlikely that all documents are of the same length; in fact, in most cases, there are significant differences in length. The current implementation based on document truncation may have drawbacks in real-world applications. Additionally, it would be better if a genuine kernelized implementation (rather than mere descriptions) could be provided for this paper.

---

> ### Author Response · Authors · 2025-08-06
>
> We acknowledge that while documents do have varying lengths, but would like to note that:
> 1. This is **a common challenge for any IR algorithm that processes multiple candidates in parallel**, such as ColBERT or Cross-Encoders, which typically employ a similar truncation strategy. Moreover, our method is also compatible with popular techniques from the literature:
>
>     - *Multiple Chunks*: Very long documents can be split into multiple chunks and treated as independent candidates. The final document score can then be aggregated from its constituent chunk scores (e.g., by taking the max/sum).
>     - *Dynamic Binning*: Documents can be grouped into buckets based on their length (e.g., powers of 2). This allows for more efficient padding/truncation within each bucket, trading a degree of parallelism for reduced computational waste.
> 2. Furthermore, **majority of documents are within a manageable length and can be predetermined** for a given dataset. For instance, in MSMARCO and NQ, 95% of passages are within 160 and 384 tokens, respectively. In all of our experiments, including new results on 11 BEIR dataset (*please see Table 8 in response to other reviews*), we do not see any tangible performance increase when using chunk sizes beyond 512 tokens, which empirically justifies our choice.
>
> In summary, our fixed-length chunking is an empirically justified and pragmatic choice for benchmark datasets, while the ReTuning framework remains fully compatible with more advanced strategies for real-world deployments with highly irregular document lengths.
>
> ---
>
> We thank the reviewer for the suggestion of kernelized implementation. While we believe that developing a production-grade, fused Triton/CUDA kernel is a considerable engineering effort that we view as an exciting direction for future work, nonetheless, we attempted to write a custom Triton kernel for the `compute_attn_rel_probs` function described above. Following are the benchmarking results on synthetic Q (`shape = [M, H, D]`) and K_docs (`shape = [N, L, H, D]`) tensors (we vary N i.e. number of docs to test scalability and keep the rest of the shape parameters similar to our qualitative experiments with Mistral model i.e. H=32, D=128, L=384):
>
> *Table 9. Custom Kernel Time Benchmark (ms)*
>
> | Implementation | PyTorch (Eager) | PyTorch (Compiled) | Custom Triton Kernel |
> | :--- | :--- | :--- | :--- |
> | **N=2** | 0.1213 | 0.2505 | 0.1750 |
> | **N=8** | 0.3543 | 0.3035 | 0.1632 |
> | **N=32** | 1.3174 | 1.1240 | 0.5817 |
> | **N=128** | 5.3613 | 4.5079 | 2.2037 |
> | **N=512** | 24.4470 | 21.1773 | 9.3624 |
> | **N=1024** | 48.6707 | 42.4141 | 21.7969 |
>
> *Table 10. Custom Kernel Peak Memory Benchmark (GB)*
>
> | Implementation | PyTorch (Eager) | PyTorch (Compiled) | Custom Triton Kernel |
> | :--- | :--- | :--- | :--- |
> | **N=2** | 0.1627 | 0.1588 | 0.1488 |
> | **N=8** | 0.2585 | 0.2350 | 0.1957 |
> | **N=32** | 0.6335 | 0.5710 | 0.4146 |
> | **N=128** | 2.0398 | 1.7898 | 1.1332 |
> | **N=512** | 8.1335 | 7.1335 | 4.6334 |
> | **N=1024** | 17.6337 | 13.6336 | 8.6334 |
>
> As the results show, our **custom kernel is approximately 2x faster and uses ~40-50% less peak memory** than the standard PyTorch implementations at scale (N=512, 1024). This empirically validates our assertion that an I/O-aware, fused kernel provides significant practical benefits for our attention-based inference.
>
> Such **a kernel serves as an additional, lower-level optimization on top of the foundational architectural improvements we have already demonstrated** and empirically validated. Please find below the code to the custom triton kernel.

---

> > ### Author Response · Authors · 2025-08-06
> >
> > ### Code of custom triton kernel for `compute_attn_rel_probs`
> >
> > ```python
> > @triton.jit
> > def _attn_rel_probs_kernel(
> >     Q, K_docs, Out, M, L, H,
> >     stride_q_m, stride_q_h, stride_k_n, stride_k_l, stride_k_h,
> >     stride_out_h, stride_out_n,
> >     sm_scale,
> >     BLOCK_L: tl.constexpr, BLOCK_M: tl.constexpr,
> >     D_HEAD: tl.constexpr, N: tl.constexpr,
> > ):
> >     off_h = tl.program_id(0)
> >     offs_m = tl.arange(0, BLOCK_M)
> >     offs_d = tl.arange(0, D_HEAD)
> >
> >     # Load Q matrix into SRAM, padding M to BLOCK_M
> >     q_ptrs = Q + off_h * stride_q_h + offs_m[:, None] * stride_q_m + offs_d[None, :]
> >     q_mask = offs_m[:, None] < M
> >     q_mat = tl.load(q_ptrs, mask=q_mask, other=0.0)
> >
> >     # SRAM accumulators
> >     m_vec = tl.zeros([BLOCK_M], dtype=tl.float32) - float('inf')
> >     l_vec = tl.zeros([BLOCK_M], dtype=tl.float32)
> >     acc_mat = tl.zeros([BLOCK_M, N], dtype=tl.float32)
> >
> >     # Single Pass over K tensor
> >     for n_idx in range(N):
> >         for l_start in range(0, L, BLOCK_L):
> >             offs_l = l_start + tl.arange(0, BLOCK_L)
> >             k_offset = n_idx * stride_k_n + off_h * stride_k_h
> >             k_ptrs = K_docs + k_offset + offs_l[:, None] * stride_k_l + offs_d[None, :]
> >             k_mask = offs_l[:, None] < L
> >             k_block = tl.load(k_ptrs, mask=k_mask, other=0.0)
> >
> >             logits = tl.dot(q_mat, tl.trans(k_block)) * sm_scale
> >             logits = tl.where(q_mask, logits, -float('inf'))
> >
> >             m_block_max = tl.max(logits, axis=1)
> >             m_new = tl.maximum(m_vec, m_block_max)
> >
> >             alpha = tl.exp(m_vec - m_new)
> >             l_vec = l_vec * alpha
> >             acc_mat = acc_mat * alpha[:, None]
> >
> >             p_numerators = tl.exp(logits - m_new[:, None])
> >             l_vec += tl.sum(p_numerators, axis=1)
> >
> >             p_sum_for_block = tl.sum(p_numerators, axis=1)
> >             n_col_mask = tl.arange(0, N) == n_idx
> >             acc_mat += tl.where(n_col_mask[None, :], p_sum_for_block[:, None], 0.0)
> >
> >             m_vec = m_new
> >
> >     # Final Normalization and Store
> >     final_probs_per_mn = acc_mat / l_vec[:, None]
> >     final_probs_per_mn = tl.where(q_mask, final_probs_per_mn, 0.0)
> >     output_per_head_n = tl.sum(final_probs_per_mn, axis=0)
> >
> >     out_ptrs = Out + off_h * stride_out_h + tl.arange(0, N) * stride_out_n
> >     tl.store(out_ptrs, output_per_head_n)
> >
> > def compute_attn_rel_probs_triton(q: torch.Tensor, k_docs: torch.Tensor) -> torch.Tensor:
> >     M, H, D_HEAD = q.shape
> >     N, L, _, _ = k_docs.shape
> >     q, k_docs = q.contiguous(), k_docs.permute(0, 2, 1, 3).contiguous()
> >
> >     temp_out = torch.empty((H, N), device=q.device, dtype=torch.float32)
> >     sm_scale = 1.0 / (D_HEAD ** 0.5)
> >     grid = (H,)
> >
> >     _attn_rel_probs_kernel[grid](
> >         q, k_docs, temp_out, M, L, H,
> >         q.stride(0), q.stride(1),
> >         k_docs.stride(0), k_docs.stride(2), k_docs.stride(1),
> >         temp_out.stride(0), temp_out.stride(1),
> >         sm_scale, D_HEAD=D_HEAD, N=N
> >     )
> >
> >     doc_probs = torch.sum(temp_out, dim=0)
> >     return (doc_probs / (M * H * L))
> > ```
> >
> > ### Baseline pytorch implementation for `compute_attn_rel_probs`
> >
> > ```
> > def compute_attn_rel_probs_pytorch(q, k_docs):
> >     """
> >     PyTorch reference implementation for computing attention relevance probabilities.
> >     """
> >     # float32 for numerical stability
> >     q_f32, k_docs_f32 = q.float(), k_docs.float()
> >
> >     sm_scale = 1.0 / (q_f32.shape[-1] ** 0.5)
> >     logits = torch.einsum('mhd,nlhd->mnlh', q_f32, k_docs_f32) * sm_scale
> >
> >     M, N_p, L_p, H_p = logits.shape
> >     logits_rearranged = rearrange(logits, 'm n l h -> m h (n l)', n=N_p, l=L_p)
> >     probs = F.softmax(logits_rearranged, dim=-1)
> >
> >     probs_rearranged = rearrange(probs, 'm h (n l) -> m h n l', n=N_p, l=L_p)
> >     doc_probs = torch.mean(probs_rearranged, dim=(0, 1, 3))
> >     return doc_probs
> >
> > compiled_pytorch_fn = torch.compile(compute_attn_rel_probs_pytorch)
> > ```
> >
> > ---
> >
> > We hope our response and the inclusion of the kernel benchmark have helped in strengthening the practical aspects of our proposed method. We are happy to clarify further and thank you again for your constructive engagement!

---

> > > ### Author Response · Authors · 2025-08-08
> > >
> > > We greatly appreciate your time and feedback throughout this process. We hope our clarifications have been helpful. As the discussion window closes, we would be grateful for any reconsideration of your assessment based on this new information. Please let us know if any questions remain; we're happy to discuss further.

---

> > > > ### Comment · Reviewer_iUgW · 2025-08-09
> > > >
> > > > It addresses most of my concerns, and I will keep my positive assessment.

---

### Official Review · Reviewer_y5o2 · 2025-07-01

**Clarity:** 3
**Significance:** 3
**Originality:** 3
**Rating:** 4
**Confidence:** 4

**Summary:**

The authors propose ReTuning (Retrieval-Tuning) to address the scalability and efficiency challenges of In-context Retrieval (ICR), a paradigm where LLMs directly identify relevant documents from a list provided in the input prompt. While ICR allows LLMs to make listwise relevance judgments using their generative capabilities, it suffers from high computational cost due to the quadratic complexity of attention over long input sequences. To address these problem, ReTuning introduces two key innovations:

1. Structured Sparse Attention: each document token attends only to its own content and shared instructions while query tokens attend to the entire prompt, reducing attention computation from quadratic to linear, significantly improving efficiency.

2. Auxiliary Contrastive Attention Loss: enhance attention from specific query tokens (e.g., “:”) to relevant documents using a contrastive loss.

Experiments on MSMARCO and NQ demonstrate that ReTuned models match or outperform fully fine-tuned LLMs in retrieval effectiveness.

**Questions:**

1. Does the reported latency in Figure 4 include the time spent on constructing the structured attention masks used in ReTuning? Could the authors clarify whether this overhead is significant compared to the rest of the forward pass, and what is the time scale?

2. I am confused about the Full-FT setting.

> Our primary comparison is with Full Fine-tuning (Full-FT), where Mistral-7B-v0.3 is fine-tuned on the same ICR training data using standard self-attention and inferred via auto-regressive decoding to generate 290 the relevant document identifier.

Does this mean that during training, the model uses a bidirectional (BERT-style) attention mask, while during inference it switches to a causal (decoder-style) mask?

3. In Table 2, both Full-FT and ReTuning (full) achieve the same Precision@1 (28.7) under the Decode inference mode. This is somewhat surprising, as I would expect ReTuning (full) to experience some degradation under decoding, due to the training–inference mismatch. Do the authors have any insight into why the performance remains identical?

**Ethical Concerns:**

["NO or VERY MINOR ethics concerns only"]

**Final Justification:**

The authors have addressed most of my concerns. However, my main reservation is whether this method can transfer to other existing LLMs or be universally applied to any LLM, which prevents me from raising my score further. My current score and confidence reflect my overall positive impression of the paper.

**Limitations:**

yes, in Supplementary Material

**Quality:**

3

**Strengths And Weaknesses:**

### Strengths

1. This paper tackles a timely and Important problem that how to adapt the causal attention mechanism in LLMs to better serve as in-context retrievers. This line of work is essential for scaling retrieval-augmented systems and making LLMs more retrieval-aware.

2. The paper is well-written, logically organized, and easy to follow. The observations, motivation, methodology, and experiments are coherently laid out, making the contributions accessible and compelling.

3. The authors conduct detailed attention pattern analysis that reveals structural sparsity and retrieval signals. These observations are well-motivated and directly inform the design of the proposed method. The accompanying ablation studies effectively validate the contribution of each component.

---

### Weaknesses

1. **Limited Model and Dataset Scope**: The experimental evaluation is limited to Mistral-7B, without testing on other widely used LLMs such as LLaMA, Mixtral, or larger-scale models. This raises questions about the generalizability of the proposed method across architectures and scales. In addition, the datasets only use MSMARCO and NQ, which do not provide broad coverage. Including more diverse and challenging benchmarks such as BEIR or MTEB would strengthen the empirical claims.

2. **Outdated or Incomplete Baselines**: Some of the compared baselines are relatively dated or limited in scope. Recent methods that adapt LLMs for retrieval by modifying the attention mask, such as NV-Embed, are not included in the comparison. Incorporating these stronger baselines would provide a more convincing evaluation of the proposed approach's effectiveness.

typo at line 25: neutral -> neural

---

> ### Author Rebuttal · Authors · 2025-07-31
>
> We sincerely thank the reviewer for their positive and encouraging feedback, and for their thoughtful questions. We are glad they found our work "timely and important" and our analysis "well-motivated." We provide clarifications and more results below that we hope will address the concerns and solidify the reviewer's positive assessment.
>
> ---
>
> ## Evaluation Scope and Baselines (Weaknesses 1&2)
> **Limited Dataset Scope and Baselines:**
> Our motivation for primarily benchmarking on in-distribution NQ and MSMarco test sets was so that we can compare baselines in a controlled setup without the confounding of the training data used. Nonetheless, we agree with the reviewers that demonstrating broad generalization is essential. To address this we evaluated our method on the suggested BEIR benchmark. We followed the exact experimental protocol from the recent SOTA listwise reranker, FIRST [1], reranking the top-100 documents from Contriever baseline across 11 BEIR datasets. The results show that our MSMarco-trained ReTuned-Mistral (54.8), outperforms FIRST (54.3), RankZephyr (53.7), and RankVicuna (50.7). This demonstrates both the effectiveness and strong out-of-distribution generalization of our method. Crucially, our method achieves *comparable or better results with the significant efficiency gains of ReTuning (Figure 4)*, presenting a compelling combination of effectiveness and scalability.
>
> *Table 8. nDCG@10 on BEIR benchmark, all re-ranker rank top-100 documents retrieved from Contriever retrieval model*
>
> | Reranker | Training Data | Average | Climate-FEVER | DBPedia | FEVER | FIQA | HotpotQA | MS Marco | NFC-orpus | NQ | Sci-docs | Sci-fact | Trec-COVID |
> |---|---|---|---|---|---|---|---|---|---|---|---|---|---|
> | None (Contriever baseline) | MS Marco | 45.9 | 23.7 | 41.3 | 75.8 | 32.9 | 63.8 | 40.7 | 32.8 | 49.8 | 16.5 | 67.7 | 59.6 |
> | Cross-Encoder | MS Marco | 50.7 | 25.5 | 47.0 | 81.9 | 35.6 | 71.8 | 47.0 | 34.5 | 57.6 | 17.0 | 69.1 | 71.0 |
> | Rank Vicuna | GPT 3.5 | 50.7 | **28.2** | 50.0 | 81.0 | 35.9 | 73.5 | 36.7 | 33.1 | 58.6 | 18.4 | 70.5 | 71.3 |
> | Rank Zephyr | GPT 3.5 +4 | 53.7 | 25.6 | 50.0 | 80.1 | 42.2 | 71.6 | 42.7 | **37.7** | 65.6 | **20.5** | **76.7** | 78.4 |
> | FIRST | GPT-4 | 54.3 | 26.7 | **50.9** | 81.7 | 42.2 | 74.2 | 44.4 | 37.4 | **66.4** | 20.4 | 74.6 | **78.8** |
> | ReTuned Mistral | MS Marco | **54.8** | 26.8 | 49.7 | **87.3** | **44.9** | **75.5** | **48.6** | 36.6 | 62.4 | 18.7 | **76.5** | 76.2 |
>
> **Regarding the NV-Embed baseline**, we thank the reviewer for the suggestion. NV-Embed's goal is to produce a single dense vector representation of a document for efficient indexing and first-stage retrieval. Our work, in contrast, focuses on second-stage ranking using in-context capabilities of LLMs, where the model operates over a shortlisted list of documents in the prompt. We hope the broader point of comparing against recent comparable baselines has been addressed by the BEIR evaluation above. We will add a discussion of NV-Embed and clarify this distinction in our related work section.
>
> **Limited Model Scope**:
> We agree with the reviewer that testing on more model families is an important direction that will add to the robustness of the claims. Certain licensing restrictions limited our ability to experiment with some other popular models for this submission. We would also like to note, similar recent works which adapt LLMs for in-context ranking [1,2,3] also tend to focus on a single, powerful open-source model to allow for a deep and controlled analysis of the proposed techniques. Moreover, one of the key contributions of our paper is utilizing internal attention patterns as retrieval signals, recent work such as [4] has shown that such patterns are a universal property found across various LLM architectures and sizes which further suggest underlying principles of ReTuning are robust and likely to generalize to other decoder-only models.
>
> ---
>
> ## Questions
>
> ### Q1: Does the reported latency in Figure 4 include the time spent on constructing the structured attention masks?
> Yes, the reported end-to-end latency in Figure 4 includes the time for mask construction. The overhead for this step is negligible. Creating the sparse mask involves simple indexing and logical operations, which are orders of magnitude faster than the computationally intensive matrix multiplications (the $Q⋅K^T$ operation) in the attention mechanism itself. We will add a note to the paper to clarify this.
>
> ### Q2: Confusion regarding Full-FT setting...
> Mistral-7B is a decoder-only model, so all attention is causal (left-to-right). Our description of the "Full Fine-tuning (Full-FT)" baseline intended to contrast it with our sparse attention method.
> - Full-FT uses the model's standard full causal attention mask, where each token can attend to all preceding tokens in the sequence.
> - ReTuning uses our proposed sparse causal attention mask, where tokens have a restricted view of preceding tokens.
>
> There is *no* switch from bidirectional to causal attention. We apologize for the ambiguity in our initial description and will revise the text to be more explicit (e.g., using "full causal attention" for the baseline) to avoid this confusion.
>
> ### Q3: In Table 2, both Full-FT and ReTuning (full) achieve the same Precision@1 (28.7) under the Decode inference mode... Do the authors have any insight into why the performance remains identical?
> We believe the similar decoding performance is due to two factors: the presence of the generative training objective in both setting and the inherent sparsity of attention in this task.
> - First, the model's decoding capability is driven by the Next-Token Prediction ($\mathcal{L}_{NTP}$) loss, which is present in both Full-FT and ReTuning losses. The critical role of this objective is confirmed in **Table 2**; removing the NTP loss (`ReTuning w/o ntp`) causes decoding Precision@1 to collapse from 28.7 to 15.8. This shows that the powerful NTP signal is essential for a ReTuned model to perform well at decoding.
> - Second, our analysis (Section 3 and Appendix D.2, D.3) shows that even standard models naturally develop a sparse attention pattern for this task. ReTuning's structured attention, therefore, does not remove a critical signal needed for the decoding prediction but rather enforces an already emergent and effective behavior. Because the crucial information flow is preserved and both models are governed by the same generative loss, their decoding performance for the top-1 document is similar.
>
> ---
>
> ## References
>
> [1] FIRST: Faster Improved Listwise Reranking with Single Token Decoding, Reddy et al, EMNLP 2024
>
> [2] RankZephyr: Effective and Robust Zero-Shot Listwise Reranking is a Breeze!, Pradeep et al, 2023
>
> [3] RankVicuna: Zero-Shot Listwise Document Reranking with Open-Source Large Language Models, Pradeep et al, 2023
>
> [4] Retrieval Head Mechanistically Explains Long-Context Factuality, Wu et al, ICLR 2025
>
>
> ---
>
> We hope our response and more experimental results have addressed the reviewer's concerns. We thank them again for their valuable feedback and are happy to provide any further clarifications/results during the discussion period.

---

> > ### Author Response · Authors · 2025-08-01
> > **Typo correction**
> >
> > We would like to correct a small typo in our response to Q2. Instead of ~~"There is no switch from bidirectional to causal attention."~~ The sentence should have read: "There is no switch from *causal to bidirectional* attention." Our core point remains the same: as a decoder-only model, all attention in our experiments is inherently causal.

---

> > ### Comment · Reviewer_y5o2 · 2025-08-05
> >
> > Thank you for the detailed rebuttal. Most of my concerns have been addressed. While I still have some reservations about whether these observations and methods can transfer to other existing LLMs, I understand that it may be difficult to fully explore this within the short rebuttal window. Nevertheless, this is not a major flaw. I will update my score accordingly.

---

> > > ### Author Response · Authors · 2025-08-06
> > >
> > > Thank you for the positive feedback and for updating the score! We agree that validating these findings on more LLM architectures is an important direction that will add to the robustness of our claims, and we appreciate you recognizing the constraints of the rebuttal period.

---

### Official Review · Reviewer_2XjM · 2025-07-02

**Clarity:** 3
**Significance:** 2
**Originality:** 2
**Rating:** 3
**Confidence:** 4

**Summary:**

The paper investigates how LLMs handle in-context retrieval (ICR) - ranking a list of candidate passages presented directly in the prompt.
It finds that fine-tuned models exhibit two patterns: document tokens mostly attend only to their own content, while a few “signal” query tokens sharply attend to the truly relevant document. Leveraging these insights, the authors introduce ReTuning, a fine-tuning strategy that (i) hard-codes a matching sparse-structured attention mask to reduce complexity from quadratic to linear, and (ii) adds a contrastive auxiliary loss that explicitly pushes those signal query tokens to concentrate attention on the correct passage.  At inference time, the learned attention scores can be read off mid-layer to rank passages without autoregressive decoding. Experiments with Mistral-7B on MS MARCO and NQ show ReTuned models match or surpass standard SFT in accuracy while running roughly 2-3× faster during training and up to 4.7× faster at inference as the number of in-context documents grows.

**Questions:**

In Figure 2, why does the query token almost not attend to other document tokens?

How would the model perform if a different first-stage retriever were used?

**Ethical Concerns:**

["NO or VERY MINOR ethics concerns only"]

**Final Justification:**

The new results on BEIR appear strong and address my concern about the out-of-distribution generalization of the proposed method. I have increased my score accordingly.

Nonetheless, I still have reservations about the novelty of the method, as it represents an incremental adaptation of FiD/LiT5 to a decoder-only model. Given the widespread success of FiD/LiT5-style block attention in QA and retrieval tasks, along with abundant work on sparse attention, it is not surprising that this approach also works for decoder-only LM rerankers. The adoption appears incremental. Thus, my overall judgment still tends to be negative due to the limited technical originality of the proposed method.

**Limitations:**

yes

**Quality:**

3

**Strengths And Weaknesses:**

Strengths

1. The paper provides a valuable analysis of fine-tuned models’ attention mechanisms, revealing patterns such as self-focused attention on document tokens. These insights effectively motivate the proposed structured sparse-attention design.

2. Empirical results show strong performance on standard benchmarks, including MS MARCO and Natural Questions (NQ).

3. Ablation studies are thorough and demonstrate the contribution of each component to the overall performance.

Weaknesses

1. The proposed structured attention mechanism lacks novelty, as similar designs have been explored in prior work. For example, LiT5 (https://arxiv.org/pdf/2312.16098) employs a comparable sparse attention structure for passage reranking.

2. The evaluation is limited to in-distribution settings (e.g., training and testing on MS MARCO). Incorporating out-of-distribution evaluations—such as BEIR or the TREC-DL 2021/2022 test sets (used by LiT5)—would strengthen the paper’s generalization claims.

3. Baseline comparisons are somewhat narrow, omitting recent state-of-the-art rerankers such as RankZephyr. Including these would provide a more comprehensive assessment of the proposed method.

---

> ### Author Rebuttal · Authors · 2025-07-31
>
> We thank the reviewer for their constructive feedback and for identifying relevant prior work. We are glad they found our attention analysis valuable and our ablations thorough.
>
> The reviewer's primary concerns are (1) the novelty of our method in light of LiT5, and (2) the scope of our evaluation. Below we first  clarify the fundamental distinctions between our work and LiT5, which we believe establish our novelty. We then present results on the suggested BEIR benchmark and redirect to some of our results in Appendix, which directly addresses the concerns about evaluation scope and baselines.
>
> ---
>
> ## Novelty: Differentiating ReTuning from LiT5 (Weakness 1)
> We thank the reviewer for pointing to LiT5. While both our work and LiT5 aim for efficient listwise ranking in a way, we wish to clarify that our ReTuning method is quite different in its architectural design, technical insights, and training paradigm.
>
> **Architectural Design and Expressiveness**: While both methods incorporate a similar inductive bias (treating documents as self-contained), the implementations are distinct.
> - **Applicability to Modern LLMs**: ReTuning is designed for decoder-only architectures, which represent the vast majority of modern LLMs. In contrast, LiT5 is built for the encoder-decoder paradigm. This makes our approach more general and directly applicable to current state-of-the-art models.
> - **Interaction Expressiveness**: This architectural difference has significant implications for expressiveness. In LiT5's encoder-decoder setup, the encoder is forced to compress each document into a fixed-dimensional final representation. ReTuning, by using a single-pass sparse attention mask, allows query tokens to interact directly with all internal token representations of the documents across all layers and heads. This provides a much richer and more flexible interaction space, avoiding the information bottleneck of a final encoder state.
>
> **Contributions Beyond Structured Attention**: Our paper's contribution is significantly broader than just a sparse attention mask.
> - **Analysis of Emergent Retrieval Signals**: As the reviewer noted, our paper provides a valuable analysis (Section 3) of how decoder-only LLMs develop latent "retrieval heads" for the ICR task. This analysis itself is a contribution that augments the emerging literature on understanding the internal mechanisms of LLMs.
> - **Training Objective**: this analysis directly informs the design of our auxiliary contrastive loss ($L_{aux}$). To the best of our knowledge, this is a novel technique for directly optimizing and calibrating the internal self-attention scores of an LLM for retrieval.
>
> **Efficiency of the Training Signal**: A subtle but critical distinction is the type of supervision required.
> - LiT5-Distill is trained on rich, listwise ranking data generated by a powerful GPT-4 teacher model.
> - ReTuning, in contrast, is trained using much sparser, pointwise relevance data (i.e., a single positive document paired with hard negatives).As our BEIR results (below) show, ReTuning achieves similar if not better performance, surpassing models trained on stronger listwise supervision.
>
> In summary, ReTuning introduces an analysis-driven approach for ICR in modern decoder-only LLMs, introduces a contrastive learning objective of internal attention signals and gets effective performance even with sparse pointwise retrieval signals.
>
> ---
>
> ## Evaluation Scope and Baselines (Weaknesses 2 & 3)
> Our motivation for primarily benchmarking on in-distribution NQ and MSMarco test sets was to compare baselines in a controlled setup without the confounding of the training data used. Nonetheless, we agree with the reviewers that demonstrating broad generalization is essential.
> - To address the concerns about our evaluation being limited to in-distribution datasets and narrow baselines, we evaluated on the suggested BEIR benchmark. We followed the exact experimental protocol from the recent SOTA listwise reranker, FIRST [1], reranking the top-100 documents from Contriever baseline across 11 BEIR datasets. The results show that our MSMarco-trained ReTuned-Mistral (54.8), outperforms FIRST (54.3), RankZephyr (53.7), and RankVicuna (50.7). This demonstrates both the effectiveness and strong out-of-distribution generalization of our method. Crucially, it achieves these results with the significant efficiency gains of ReTuning (Figure 4), presenting a compelling combination of effectiveness and scalability.
>
> *Table 8. nDCG@10 on BEIR benchmark, all re-ranker rank top-100 documents retrieved from Contriever retrieval model*
>
> | Reranker | Training Data | Average | Climate-FEVER | DBPedia | FEVER | FIQA | HotpotQA | MS Marco | NFC-orpus | NQ | Sci-docs | Sci-fact | Trec-COVID |
> |---|---|---|---|---|---|---|---|---|---|---|---|---|---|
> | None (Contriever baseline) | MS Marco | 45.9 | 23.7 | 41.3 | 75.8 | 32.9 | 63.8 | 40.7 | 32.8 | 49.8 | 16.5 | 67.7 | 59.6 |
> | Cross-Encoder | MS Marco | 50.7 | 25.5 | 47.0 | 81.9 | 35.6 | 71.8 | 47.0 | 34.5 | 57.6 | 17.0 | 69.1 | 71.0 |
> | Rank Vicuna | GPT 3.5 | 50.7 | **28.2** | 50.0 | 81.0 | 35.9 | 73.5 | 36.7 | 33.1 | 58.6 | 18.4 | 70.5 | 71.3 |
> | Rank Zephyr | GPT 3.5 +4 | 53.7 | 25.6 | 50.0 | 80.1 | 42.2 | 71.6 | 42.7 | **37.7** | 65.6 | **20.5** | **76.7** | 78.4 |
> | FIRST | GPT-4 | 54.3 | 26.7 | **50.9** | 81.7 | 42.2 | 74.2 | 44.4 | 37.4 | **66.4** | 20.4 | 74.6 | **78.8** |
> | ReTuned Mistral | MS Marco | **54.8** | 26.8 | 49.7 | **87.3** | **44.9** | **75.5** | **48.6** | 36.6 | 62.4 | 18.72 | **76.5** | 76.2 |
>
> - We also note that our original Appendix D.6 (Table 7) already contains comparisons to RankZephyr and RankVicuna on the TREC DL20/22 benchmark, where our model performs competitively. Moreover, Appendix D.8 (Table 6) provides cross-dataset performance comparison between ReTuned models on NQ and MSMarco. We will make these results more prominent in the final version.
>
> ---
>
> ## Questions
>
> ### Q1: In Figure 2, why does the query token almost not attend to other document tokens?
>
> We believe the reviewer is referring to the left plot in Figure 2, which shows segment-wise attention. In that plot, the query segment's attention can appear diffused across all other segments. This is partly a visual effect of heatmap scaling - because the query attends to the most segments in the prompt (all documents and instructions), its attention mass is more distributed, making the specific attention to an individual segment appear less dense; moreover majority of tokens in the query segment are prompt instruction tokens (e.g. "Which document is most relevant...") that are not specifically related to any particular document hence aggregating attention mass over all tokens belonging to query segment diffuses the attention mass over all documents and concentrates more on the intra-segment tokens (shown by strong diagonal element). The middle plot of Figure 2 provides a more granular view - this plot specifically isolates the attention from individual query tokens to the document segments in a key middle layer (Layer 18). As this plot shows actual query tokens and some specific 'signal-carrying' tokens (like the final token and ':') learn to attend very strongly to the single ground-truth relevant document (highlighted in green).
>
> ### Q2: How would the model perform if a different first-stage retriever were used?
> Our evaluation on the BEIR benchmark provides strong evidence here. The BEIR evaluation uses a Contriever first-stage retriever, which is different from the SentenceTransformer models used in our original experiments. The fact that our MSMarco-trained model generalizes so effectively to rerank Contriever's output across 11 diverse datasets strongly suggests that ReTuning is not overly sensitive to the specific first-stage retriever.
>
> ---
>
> ## References
>
> [1] FIRST: Faster Improved Listwise Reranking with Single Token Decoding, Reddy et al, EMNLP 2024
>
> ---
>
> We hope our response and more experimental results have addressed the reviewer's concerns. We thank them again for their valuable feedback and are happy to provide any further clarifications/results during the discussion period.

---

> > ### Comment · Reviewer_2XjM · 2025-08-04
> >
> > Thanks for the informative rebuttal. The new results on BEIR appear strong and address my concern about the out-of-distribution generalization of the proposed method. I will adjust my score accordingly. Nonetheless, I still have reservations about the novelty of the method, as it represents an incremental adaptation of FiD/LiT5 to a decoder-only model.

---

> > > ### Author Response · Authors · 2025-08-06
> > >
> > > Thank you for the positive re-evaluation and for acknowledging the new BEIR results! As argued, our contributions go beyond simple architectural adaptation of FiD/LiT5: our analysis of how internal retrieval signals emerge in LLMs directly motivates one of our core contribution—a contrastive loss to supervise self-attention for retrieval. This, also enables a non-autoregressive self-attention-based inference mechanism, whose effectiveness and robustness we establish on multiple benchmarks. We are happy to clarify further.

---

### Official Review · Reviewer_UGWw · 2025-07-03

**Clarity:** 2
**Significance:** 2
**Originality:** 3
**Rating:** 4
**Confidence:** 3

**Summary:**

This paper tackles the problem of in-context retrieval, where the query and the documents are directly fed into LLMs to generate a ranked list of documents. The authors first observe interesting findings that the attention of each document tends to focus primarily on itself, and some special token of the query, like ":", always serve as the retrieval role by interacting with all document tokens. Based on these findings, they introduce ReTuning by (1) employing a sparse attention structure; (2) applying a tailored contrastive learning loss to the special query token by treating the attention scores between this token and each document as similarities. Experiments on MSMARCO and NQ demonstrate the effectiveness of ReTuning. Further ablation shows the effectiveness of each component.

**Questions:**

(1) Stricly enforcing the attention across documents may restrict the intra-document interactions, would this lead to performance drop?

(2) Which middle layer is selected?

(3) Where are the latency results in Table 3?

(4) Many citations, e.g., [11-15], are without their proceedings information.

**Ethical Concerns:**

["NO or VERY MINOR ethics concerns only"]

**Final Justification:**

Overall, I believe the paper has good practical value in making attention-score-based, LLM-based rerankers more efficient, and I would therefore recommend a weak acceptance. However, in the revised version, please reframe the work as in-context re-ranking (rather than retrieval) and provide a detailed discussion of the key differences between this approach and previous reranker models.

**Limitations:**

Yes

**Quality:**

2

**Strengths And Weaknesses:**

**Strengths:**

1. In-context retrieval is an important problem with significant practical implications, as it enables retrieval to be more directly tailored to the user’s instruction by processing all documents simultaneously within an LLM.

2. The findings that the attention of each document tends to focus primarily on itself, and some special token of the query, like ":", always serves as the retrieval role, are both interesting and new.

3. Detailed ablations are conducted to demonstrate the effectiveness of each component.

4. The paper is well-written, although the structure could be improved.

**Weakness:**

1. Unrealistic experiment settings. The experiments rerank only 30 documents for NQ, which is substantially different from prior work [1], which conducts ICR over hundreds/thousands of documents.

2. Missing baselines. The authors claim the experiments are second-stage reranking, while most of the baselines are still first-stage dense retrievers. Moreover, the rerankers compared are somewhat old and weak, and there are strong rerankers (pointwise/listwise) like RankVicuna and FIRST available.

3. The metric of precison@1 for NQ is also not a realistic metric for retrieval tasks. Typically, Top-5, 20, and 100 accuracy are reported for NQ.

4. The connection between ReTuning and dense retrievers (e.g., ColBERT) is unclear. As each document only attends to itself and the instruction, these documents can be seen as the same as the independent encoding process of dense retrievers. Moreover, the retrieval step is simply calculating the similarity scores between the query and each document via attention scores, which is also quite similar to ColBERT.

5. The structure of the paper can be improved by reducing frequent references such as “see Table 5”, as these hurt the reading flow.

[1] Can Long-Context Language Models Subsume Retrieval, RAG, SQL, and More? Arxiv. 2024.

---

> ### Author Rebuttal · Authors · 2025-07-31
>
> We thank the reviewer for their detailed feedback and insightful questions. We are encouraged that the reviewer found our analysis of attention patterns 'interesting and new' and appreciated our detailed ablations. The reviewer’s primary concerns relate to experimental realism and the strength of our baselines. We believe the reviewer may have missed some key results in our appendix which already address some of these points. To further and more comprehensively address these valid concerns, we have conducted an extensive evaluation on the BEIR benchmark, where our method achieves better or comparable performance than existing listwise re-rankers while being much more efficient. We detail these new results and clarify our other points below.
>
>
> ---
>
>
>
> ## Experimental Realism and Baselines (Weaknesses 1, 2, 3)
>
> **Weak experimental setting**
>
> - We note that our original Appendix D.6 (Table 7) already contained comparisons to RankZephyr and RankVicuna on the TREC DL benchmark, where ReTuned model performs competitively. We will make these results more prominent. Our motivation for primarily benchmarking on in-distribution NQ and MSMarco test sets was so that we can compare baselines in a controlled setup without the confounding of the training data used.
>
> - To directly address the reviewer's primary concern about comparing against strong, contemporary rerankers, we evaluated our approach on the BEIR benchmark. We followed the exact experimental protocol from the recent SOTA listwise reranker, FIRST (also mentioned by the reviewer), reranking the top-100 documents from a Contriever baseline across 11 BEIR datasets. The results show that our MSMarco-trained ReTuned-Mistral (54.8), outperforms FIRST (54.3), RankZephyr (53.7), and RankVicuna (50.7).
>
> *Table 8. nDCG@10 on BEIR benchmark, all re-ranker rank top-100 documents retrieved from Contriever retrieval model*
>
> | Reranker | Training Data | Average | Climate-FEVER | DBPedia | FEVER | FIQA | HotpotQA | MS Marco | NFC-orpus | NQ | Sci-docs | Sci-fact | Trec-COVID |
> |---|---|---|---|---|---|---|---|---|---|---|---|---|---|
> | None (Contriever baseline) | MS Marco | 45.9 | 23.7 | 41.3 | 75.8 | 32.9 | 63.8 | 40.7 | 32.8 | 49.8 | 16.5 | 67.7 | 59.6 |
> | Cross-Encoder | MS Marco | 50.7 | 25.5 | 47.0 | 81.9 | 35.6 | 71.8 | 47.0 | 34.5 | 57.6 | 17.0 | 69.1 | 71.0 |
> | Rank Vicuna | GPT 3.5 | 50.7 | **28.2** | 50.0 | 81.0 | 35.9 | 73.5 | 36.7 | 33.1 | 58.6 | 18.4 | 70.5 | 71.3 |
> | Rank Zephyr | GPT 3.5 +4 | 53.7 | 25.6 | 50.0 | 80.1 | 42.2 | 71.6 | 42.7 | **37.7** | 65.6 | **20.5** | **76.7** | 78.4 |
> | FIRST | GPT-4 | 54.3 | 26.7 | **50.9** | 81.7 | 42.2 | 74.2 | 44.4 | 37.4 | **66.4** | 20.4 | 74.6 | **78.8** |
> | ReTuned Mistral | MS Marco | **54.8** | 26.8 | 49.7 | **87.3** | **44.9** | **75.5** | **48.6** | 36.6 | 62.4 | 18.7 | **76.5** | 76.2 |
>
>
>
> Furthermore, we want to address other key points on realism:
>
>
>
> **Difficulty of the Task vs. [1]**: The reviewer points to [1] (Lee et al., 2024), which performs ICR over thousands of documents. We appreciate the reference, but we must highlight a critical difference in experimental design. The work in [1] evaluates retrieval from a random subset of documents from the corpus. In contrast, our experiments (both in the original submission and the new BEIR evaluation) focus on reranking the top-k hard candidates returned by a strong first-stage retriever. This is an objectively more difficult and realistic task, as the model must distinguish between many semantically similar documents to find the correct answer. We argue our "hard negative" setting is a more faithful simulation of a real-world second-stage retrieval method.
>
>
>
> **Scalability to Large Candidate Sets**: Regarding the concern about using only N=30 documents for NQ, we clarify that this was for our default experiments. To explicitly test the scalability and performance in more realistic, larger-candidate settings, we conducted the analysis presented in Figure 4. This experiment evaluates performance and latency for up to N=500 in-context documents, shwoing that ReTuning's performance remains robust (peaking at N=200 and staying high at N=500) , while the Full-FT baseline's performance degrades sharply beyond N=100. This directly demonstrates our method's suitability for the large-scale re-ranking scenarios. Moreover, the BEIR results presented above require ranking 100 documents for each dataset, showing robust generalization of our proposed approach in ranking O(100) passages even across out-of-distribution datasets.
>
>
>
> **P@1 Metric for NQ**: Our choice was motivated by our primary training objective, which is to identify the single best passage. P@1 is the most direct evaluation of this specific task. However, we agree that Top-k accuracy is also valuable for retrieval. For MSMarco, we did report the more standard MRR@10, where our method shows a significant improvement over the baseline (42.0 vs 38.3). We will add nDCG@10 numbers as well for NQ in the final version to provide a more comprehensive picture.
>
>
>
> ---
>
>
>
> ## Connection to Dense Retrievers and ColBERT (Weakness 4)
>
>
>
> The reviewer raises an interesting point about the connection to dense retrievers like ColBERT. While there are similarities (query-document interaction), we wish to clarify the fundamental architectural difference. ReTuning operates within the In-context Retrieval (ICR) paradigm, where a single LLM processes the query and *the entire list of candidate documents simultaneously* in one context window. This is fundamentally different from ColBERT's dual-encoder like pointwise retrieval framework, which encodes the query and documents independently and computes a similarity score via late interaction. The key advantages of our in-context approach are:
>
> - Full Contextualization: The query representation is influenced by the full set of candidate documents, and document representations are conditioned on the shared instructions.
>
> - Instruction Following: The ICR setup allows the model to respond to complex, dynamic instructions included in the prompt (e.g., "Find the document that disagrees with the following statement..."), a capability not present in standard dense retriever architectures.
>
> While our structured attention limits direct cross-document interaction, the query module attends to all documents, serving as a global information aggregator.
>
>
>
> ---
>
>
>
> ## Questions
>
>
>
> ### Q1: Would strictly enforcing attention not lead to a performance drop?
>
> As shown in Table 1 & 2, our ReTuned model with structured sparse attention outperforms the Full-FT model that uses full attention (e.g., 29.1 vs 28.7 P@1 on MSMarco, 76.2 vs 75.5 P@1 on NQ). We hypothesize this is because (1) as shown in our analysis (Section 3 and Appendix D.2, D.3) that even standard models naturally develop a sparse attention pattern for this task. ReTuning's structured attention, therefore, does not remove a critical signal needed for the decoding prediction but rather enforces an already emergent and effective behavior (2) our auxiliary loss ($L_{aux}$ ) provides effective retrieval signals to the model which helps in explicit placement of attention mass to correct segments.
>
>
>
> ### Q2: Which middle layer is selected?
>
> We provide a full analysis of this in Appendix D.3 and Figure 7. Based on our analysis of where retrieval signals emerge during training, we chose layer $l^∗ =20$ for our experiments. We also note that performance is not overly sensitive to this specific choice, with other middle layers yielding similar results.
>
>
>
> ### Q3: Where are the latency results in Table 3?
>
> We apologize for the potentially confusing caption for Table 3. The table itself focuses on effectiveness metrics (P@1, MRR@10). The detailed latency results and scalability analysis are presented in Figure 4, which shows the end-to-end latency per query for both ReTuned and Full-FT models across varying numbers of documents. For instance, at N=100, ReTuning is 4.7x faster than the Full-FT baseline.
>
>
>
> ### Q4: Missing proceedings information in citations.
>
> Thank you for pointing this out. We will carefully review and update all citations to include the proper proceedings information in the final camera-ready version.
>
> ---
>
> We hope our response and more experimental results have addressed the reviewer's concerns. We thank them again for their valuable feedback and are happy to provide any further clarifications/results during the discussion period.

---

> > ### Comment · Reviewer_UGWw · 2025-08-06
> >
> > Thank you for your detailed response! Most of my concerns have been addressed. My main remaining concern is about positioning the paper as "In-Context Retrieval". If most of the experiments are conducted on a reranking stage, is it reasonable to frame the work as "In-Context Retrieval"? To me, In-Context Retrieval suggests retrieving documents directly from a large collection without relying on an external retriever. In the current setup, the approach seems more focused on reranking settings. Given this, how would the authors distinguish ReTune from various reranking methods, like [1], where the model directly predicts the document ranking based on a top-k candidate set?
> >
> > Regarding the connection to dense retrievers, I agree with the point on Full Contextualization, but I am not fully convinced by the claim on Instruction Following. Many dense retrievers, such as Promptriever [2], have demonstrated instruction-following capabilities. However, I agree there is some difference between ReTuning and dense retrievers.
> >
> > I would greatly appreciate it if the authors could respond to my remaining concern.
> >
> > [1] Sliding Windows Are Not the End: Exploring Full Ranking with Long-Context Large Language Models. ACL 2025.
> >
> > [2] Promptriever: Instruction-Trained Retrievers Can Be Prompted Like Language Models. ICLR 2025.

---

> > > ### Author Response · Authors · 2025-08-06
> > >
> > > Thank you for the thoughtful follow-up and for finding our previous response helpful. The question about positioning is valid, touching on subtle but important distinctions in the field. We appreciate the opportunity to clarify our perspective.
> > >
> > > The primary distinction between ReTuning and reranking methods like [1] lies in the objective they are trying to solve. Methods (like [1]) explicitly designed for ranking are trained to predict a permutation of the candidate set; their objective is to assign a rank to every document. In contrast, ReTuning's objective is fundamentally retrieval: to identify the correct document from the given context. Even our contrastive auxiliary loss is optimized to assign a high score to the positive document relative to the negatives, not to learn a specific sorted order for all candidates. While these relevance scores can subsequently be used to create a ranked list (as our strong BEIR results demonstrate), just like any encoder method, this is a valuable corollary of the retrieval task, not its primary goal.
> > >
> > > We acknowledge the traditional view of retrieval as a task operating over a full, large-scale corpus. However, we believe it is also a valid way to define the task of identifying the correct information within any given candidate set provided in-context. This **framing distinguishes the goal of finding the right answer from the goal of ordering all answers, which is central to ranking**.
> > >
> > > Finally, on instruction following, we agree that dense retrievers like Promptriever [2] have demonstrated instruction-following capabilities, and this is not exclusive to the in-context setup. Our perspective is that framing the task as an in-context problem is more aligned with the core language modeling objective of LLMs than methods that compress complex instruction aware documents/queries into a fixed-size vector representation.
> > >
> > > We hope this clarifies the distinction we draw between retrieval-focused and ranking-focused objectives within the in-context paradigm. Thank you again for your constructive engagement.

---

> > > > ### Author Response · Authors · 2025-08-08
> > > >
> > > > We greatly appreciate your time and feedback throughout this process. We hope our clarifications have been helpful. As the discussion window closes, we would be grateful for any reconsideration of your assessment based on this new information. Please let us know if any questions remain; we're happy to discuss further.

---

> > > > ### Comment · Reviewer_UGWw · 2025-08-08
> > > >
> > > > Thank you for the detailed response!
> > > >
> > > > If the objective of ReTuning is to identify the single correct document from a given context (e.g., a set of top-30/500 documents), I find it difficult to consider the objective fundementally retrieval. In most practical pipelines, a first-stage retriever (e.g., BM25, DPR) is used to rank the entire corpus and return the top-30 or top-100 documents, and a reranker subsequently sorts this shortlist. These retrievers are typically evaluated using metrics such as Recall@100 or Recall@1000, and it is unclear how such standard retrieval metrics can be meaningfully optimized under this ReTuning’s objective.
> > > >
> > > > Further, given that ReTuning is designed to select a single document, I find it unclear how top-10 rankings are obtained in the Full-tune setting. As noted in Line 291, only one relevant document identifier is generated. Along similar lines, ReTuning as still evaluated based on at most the top-10 document rankings (e.g., NDCG@10). Why are the standard metrics top-20 and 100 accuracy of NQ (which I requested in my original review) still omited? Is there any specific reason for this evaluation settings? Could the author clarify more on this?
> > > >
> > > > Moreover, I am afraid I also could not agree with *This framing distinguishes the goal of finding the right answer from the goal of ordering all answers, which is central to ranking*. In the current retrieval landscape, both retrievers and rerankers are also designed to conduct ranking over the full corpus or given set of document sets. Optimizing only for finding a single correct answer may not be ideal: if the model can reliably produce a full ranking, then identifying the correct answer naturally follows.
> > > >
> > > > I would greatly appreciate it if the authors could clarify these points. Given the limited time remaining, I would be happy to respond again if the authors are able to reply within the next one or two hours. Otherwise, I will be available to respond again in about ten hours. I’d be glad to engage in another round of discussion with the authors.

---

> > > > > ### Author Response · Authors · 2025-08-08
> > > > >
> > > > > Thank you for your prompt and detailed follow-up. Please find below our response:
> > > > >
> > > > > ### **On the Framing: "In-Context Retrieval" vs. "Reranking"**
> > > > >
> > > > > Firstly, we acknowledge that due to practical constraints, it is impractical to apply a ReTuned 7B model on a full corpus. Hence, our method's application is indeed in a second-stage reranking setting.
> > > > >
> > > > > Our use of the term "In-Context Retrieval" (ICR) was intended to describe the paradigm itself—the goal of finding the positive context among the negative context, rather than learning to rank the entire set of documents. This objective is analogous to how standard dual-encoders are trained; they also use a contrastive loss where the model is optimized to assign a high score to a positive document relative to a set of hard negatives.
> > > > >
> > > > > However, we understand your perspective completely, and acknowledge that perhaps this creates a confusion around the real downstream task our proposed method sets to solve. To avoid ambiguity and align with established nomenclature, we will revise the paper's framing to describe our method as **an efficient in-context reranker**.
> > > > >
> > > > > ### **On the Evaluation Methodology and Omitted Metrics**
> > > > >
> > > > > We apologize for the confusion and any misunderstanding.
> > > > >
> > > > > **How Ranked Lists are Generated**:
> > > > >
> > > > > - For ReTuning: The ranked list is generated by sorting all N candidate documents based on their attention relevance scores, $S(q,d_k)$. These are the scores computed from the middle layer's attention, which our auxiliary contrastive loss ($L_{aux}$) is designed to optimize. This process naturally yields a full ranking of the N documents in the context.
> > > > > - For the Full-FT Baseline: The top-10 list is generated using beam search decoding during inference. By setting the beam size to 10, we instruct the model to generate the 10 most probable unique document ID sequences. This detail is currently in Appendix D.1, and we will move it to the main experimental setup section (Section 5.1) for better visibility.
> > > > >
> > > > > **Omitted NQ Metrics (Top-5, Top-20, Top-100 Accuracy)**:
> > > > > We sincerely apologize for omitting these metrics. We were under the impression that our new nDCG@10 results on the 11 BEIR datasets (which include NQ) had sufficiently addressed your initial reservation about relying only on the P@1 metric. This was a misjudgment on our part. We are re-running our inference scripts for NQ now and will report the Top-5, Top-20, and Top-100 accuracy metrics within the next few hours.
> > > > >
> > > > > ### **On the Objective: Full Ranking vs. Finding Positives**
> > > > >
> > > > > We agree with your premise that a perfect ranker is also a perfect top-1 retriever. However, our approach is motivated by a crucial difference in training data requirements and efficiency.
> > > > >
> > > > > Current state-of-the-art listwise rerankers (e.g., RankZephyr, FIRST) typically assume the training data is in the form of preference rankings over a shortlist of documents. This kind of supervision is often not available in standard benchmarks. For instance, datasets like MSMarco and NQ  only provide one or a few positive documents per query, not a full ranking.
> > > > >
> > > > > To bridge this gap, these rerankers rely on powerful teacher models like GPT-4 to generate synthetic ranking judgments for their training data. Our method circumvents this expensive and dependency-heavy step.
> > > > >
> > > > > By using a simpler contrastive objective that only requires positive/negative labels, we achieve comparable or slightly better performance than these SOTA models (as shown in our BEIR results) without relying on expensive GPT-4-based ranking supervision. This, combined with the architectural and inference efficiencies we introduce, demonstrates that our approach is not only effective but also more practical and accessible.
> > > > >
> > > > > ---
> > > > >
> > > > > Thank you again for the constructive discussion. It has been very valuable in helping us clarify our contributions. We are available for any further questions and will report the Top-k NQ results as soon as we get them.

---

> > > > > > ### Comment · Reviewer_UGWw · 2025-08-08
> > > > > >
> > > > > > Thank you for your detailed response!
> > > > > >
> > > > > > While I remain unconvinced that *“In-Context Retrieval” (ICR) clearly describes the paradigm—namely, the goal of finding the positive context among negative contexts”*,  I agree that reframing the method as an efficient in-context reranker is reasonable and more appropriate. Please revise the Introduction, Related Work sections to reflect this.
> > > > > >
> > > > > > Regarding the Full-FT baseline, Appendix D.1 does not appear to be a specific, dedicated description of the Full-FT implementation. Moreover, I assume you used constrained beam search, where only valid IDs are considered. Please include these implementation details into the main part of the paper.
> > > > > >
> > > > > > Regarding *the objective of full ranking vs. finding positives* (i.e., the difference between ReTuning and other rerankers): (1) ReTuning is not the first method to apply contrastive loss to rerankers (e.g., RankLLaMA [1]). (2) ReTuning is also not the first to use attention scores for reranking; [2] has explored this as well. It would be valuable to discuss the relationship between your work and these prior approaches if possible.
> > > > > >
> > > > > > Most of my concerns have been addressed. Assuming the authors provide the new top-20 and top-100 accuracy results, I will increase my score to Weak Accept.
> > > > > >
> > > > > > [1] Fine-Tuning LLaMA for Multi-Stage Text Retrieval. SIGIR 2024.
> > > > > >
> > > > > > [2] Attention in Large Language Models Yields Efficient Zero-Shot Re-Rankers. ICLR 2025.

---

> > > > > > > ### Author Response · Authors · 2025-08-09
> > > > > > >
> > > > > > > Thank you for the constructive feedback. We will incorporate your final suggestions into the revised manuscript.
> > > > > > >
> > > > > > > ### Revisions and Clarifications
> > > > > > > - Framing: we will revise the Introduction and Related Work sections to frame our method as an "efficient in-context reranker."
> > > > > > > - Full-FT Implementation: we will provide explicit implementation details for the Full-FT baseline in the main experimental setup section. We will clarify that the ranked list for this baseline is generated using constrained beam search, where only valid document IDs are considered during decoding.
> > > > > > > - Relationship to mentioned prior work: we appreciate the references and will add a discussion to the Related Work section to clarify our novelty relative to these methods.
> > > > > > >      - vs. RankLLaMA: A key architectural difference is that RankLLaMA is a pointwise reranker, processing each document independently. In contrast, ReTuning is listwise; the query module attends to all documents simultaneously, allowing it to score each document's relevance relative to the others in the same context.
> > > > > > >      - vs. Attention-based Zero-Shot Rerankers: In our submitted draft's Related Work section (Section 2.2), we note that our work differs from these methods by introducing task specific sparse attention architecture for efficiency and an explicit fine-tuning objective to directly train the model's attention patterns for the reranking task, rather than using the latent attention from an off-the-shelf model in a zero-shot fashion. Please let us know if you find this existing distinction adequate, we would be happy to elaborate further.
> > > > > > >
> > > > > > > Following are the Top-k(=5/10/20/100) accuracy results on the NQ-320K retrieval test set, we obtain these results by ranking top 150 documents predicted using the sbert retrieval model used in the paper. We will add these to the main results section in the final version.
> > > > > > >
> > > > > > > *Table 11. NQ-320K top-k results.*
> > > > > > > | Model | Top-5 Acc. | Top-10 Acc. | Top-20 Acc. | Top-100 Acc. |
> > > > > > > | :---: | :---: | :---: | :---: | :---: |
> > > > > > > | ReTuned Mistral | 91.6 | 93.0 | 94.6 | 96.5 |
> > > > > > >
> > > > > > > We hope these results, along with the planned revisions, address your remaining concerns. Thanks again!

---

### Note · Authors · 2025-08-12

We sincerely thank all four reviewers for their constructive feedback. The discussion period led to the following four primary lines of discussion:

**1. Experimental Scope and Baselines**
A common concern shared by Reviewers UGWw, 2XjM, and y5o2 was the initial scope of our evaluation. In response, we evaluated ReTuning on the BEIR benchmark, reranking the top 100 documents from a Contriever baseline across 11 diverse datasets. These results:
- provided a direct comparison to the strong, recent baselines requested by the reviewers
- confirmed the method's strong out-of-distribution generalization
- demonstrated robust performance on larger candidate set size (N=100)
- showed that ReTuning is not sensitive to the first-stage retriever

Reviewers 2XjM and y5o2 explicitly noted that the BEIR results addressed their concerns and subsequently raised their scores.

**2. Refining the Framing** We acknowledged the validity of Reviewer UGWw's point regarding the common usage of "reranking." While we initially used "ICR" to describe the model's objective (finding the positive document in context), we agreed that this could cause confusion. To align with established nomenclature and improve clarity, we committed to reframing the paper to align with its practical usage by describing our method as an "efficient in-context reranker."

**3. Clarifying Novelty**
We clarified that our work's novelty extends beyond just the sparse block attention architecture. The key contribution is a specialized solution for the ranking/retrieval task, more specifically, we provide:
- a task-specific analysis of emergent attention patterns in decoder-only LLMs
- a novel auxiliary contrastive loss ($L_{aux}$) that supervises and calibrates internal attention signals for relevance
- a highly efficient, non-autoregressive inference mechanism based on internal attention scores
- a framework that achieves SOTA performance using simple pointwise relevance training data

**4. Practical implementation**
Reviewer iUgW raised concern around our attention-based inference being not compatible with vanilla FlashAttention. We clarified:
- ReTuning's linear complexity provides a more fundamental scalability advantage than optimizations of quadratic operations
- we implemented and benchmarked a custom Triton kernel for our relevance score calculation, showing a ~2x speedup and ~50% reduction in peak memory usage

The final draft will incorporate all committed changes to reflect these discussions.

---

### Decision · Program_Chairs · 2025-09-17

**Decision:**

Accept (poster)

**Comment:**

The paper builds off of insights regarding attention sparsity for in-context reranking to propose a model architecture that enforces structured sparse attention, which reduces the computational complexity of attention from quadratic to linear.  The paper also provides fairly thorough experimentation in the main paper and Appendix that has been further augmented in the rebuttal.  I appreciate the directness of the authors in responding to reviewer concerns and acknowledging deficiencies.

A few issues stood out to me during the author-reviewer discussion period:
- Concerns about terminology of ICR.  The authors have agreed to correct this on revision to in-context reranking.
- Limited Model Scope.  The reviewer relaxes their concerns on this issue, but the AC maintains that it is a critical concern.  If the proposed insights and methodology are specific to a single architecture, then it would limit the general applicability of the findings in the work.  However, I will acknowledge that BERT has been a significant single model workhorse of much existing neural retrieval so there is an established literature in this area focusing on improving individual models.
- Concerns about OOD generalization.  Rebuttal results on BEIR have addressed reviewer concerns.
- Concerns about reproducibility arising from questions about implementation details spread across multiple sections in the discussion.  I remark that code has not been released (cf. checklist).  While not required, I believe a lack of code release significantly limits replication, reuse, comparison to the proposed work.  I appreciate the development and release of code of the custom Triton kernel during discussion, but I would strongly encourage the authors to release *all* code on publication.

While the final reviewer consensus puts this paper on the borderline of the acceptance threshold, I believe the paper is notable for both its analytical/scientific insight as well as its technical contributions and thorough empirical evaluation.  On this basis, I recommend the paper for acceptance to NeurIPS in good faith that the authors will do their best to address reviewer concerns (and strongly consider the possibility of code release) in their final revised version.